# On the cross-population generalizability of gene expression prediction models

Kevin L. Keys[1,2]*, Angel C. Y. Mak[1], Marquitta J. White[1], Walter L. Eckalbar[1], Andrew W. Dahl[1], Joel Mefford[1], Anna V. Mikhaylova[3], María G. Contreras[1,4], Jennifer R. Elhawary[1], Celeste Eng[1], Donglei Hu[1], Scott Huntsman[1], Sam S. Oh[1], Sandra Salazar[1], Michael A. Lenoir[5], Jimmie C. Ye[6,7], Timothy A. Thornton[3], Noah Zaitlen[8], Esteban G. Burchard[1,7‡], Christopher R. Gignoux[9,10‡]*

1 Department of Medicine, University of California, San Francisco, California, United States of America, 2 Berkeley Institute for Data Science, University of California, Berkeley, California, United States of America, 3 Department of Biostatistics, University of Washington, Seattle, Washington, United States of America, 4 San Francisco State University, San Francisco, California, United States of America, 5 Bay Area Pediatrics, Oakland, California, United States of America, 6 Department of Epidemiology and Biostatistics, University of California, San Francisco, California, United States of America, 7 Department of Bioengineering and Therapeutic Biosciences, University of California, San Francisco, California, United States of America, 8 Department of Neurology, University of California, Los Angeles, California, United States of America, 9 Colorado Center for Personalized Medicine, University of Colorado Anschutz Medical Campus, Aurora, Colorado, United States of America, 10 Department of Biostatistics and Informatics, School of Public Health, University of Colorado Anschutz Medical Campus, Aurora, Colorado, United States of America

‡ These authors are joint senior authors on this work.
* klkeys@g.ucla.edu (KLK); chris.gignoux@ucdenver.edu (CRG)

**Data Availability Statement:** Data from SAGE are available through NCBI dbGaP under accession number phs000921.v3.p1. All simulation data files and results are available online through Dryad at

## Abstract

The genetic control of gene expression is a core component of human physiology. For the past several years, transcriptome-wide association studies have leveraged large datasets of linked genotype and RNA sequencing information to create a powerful gene-based test of association that has been used in dozens of studies. While numerous discoveries have been made, the populations in the training data are overwhelmingly of European descent, and little is known about the generalizability of these models to other populations. Here, we test for cross-population generalizability of gene expression prediction models using a dataset of African American individuals with RNA-Seq data in whole blood. We find that the default models trained in large datasets such as GTEx and DGN fare poorly in African Americans, with a notable reduction in prediction accuracy when compared to European Americans. We replicate these limitations in cross-population generalizability using the five populations in the GEUVADIS dataset. Via realistic simulations of both populations and gene expression, we show that accurate cross-population generalizability of transcriptome prediction only arises when eQTL architecture is substantially shared across populations. In contrast, models with non-identical eQTLs showed patterns similar to real-world data. Therefore, generating RNA-Seq data in diverse populations is a critical step towards multi-ethnic utility of gene expression prediction.

https://doi.org/10.7272/Q6RN362Z. All source code for analyses in this manuscript is available on Github at https://github.com/asthmacollaboratory/sage-geuvadis-predixcan.

**Funding:** This work was supported in part by the Sandler Family Foundation, the American Asthma Foundation, the RWJF Amos Medical Faculty Development Program, the Harry Wm. and Diana V. Hind Distinguished Professor in Pharmaceutical Sciences II, the National Heart, Lung, and Blood Institute (NHLBI) R01HL117004, R01HL128439, R01HL135156, X01HL134589, R01HL141992, and R01HL104608, the National Human Genome Research Institute (NHGRI) U01HG007419 and U01HG009080, the National Institute of Environmental Health Sciences (NIEHS) R01ES015794 and R21ES24844, the National Institute on Minority Health and Health Disparities (NIMHD) P60MD006902 and R01MD010443, and the Tobacco-Related Disease Research Program under Award Numbers 24RT-0025, 27IR-0030. Research reported in this article was funded by the National Institutes of Health Common Fund and Office of Scientific Workforce Diversity under three linked awards RL5GM118984, TL4GM118986, and UL1GM118985 administered by the National Institute of General Medical Sciences. (K.L.K, A.C. Y.M, M.J.W, M.G.C, J.R.E, C.E, D.H, S.H, S.S.O, S. S, E.G.B). K.L.K was additionally supported by a diversity supplement of NHLBI R01HL135156, the UCSF Bakar Computational Health Sciences Institute, the Gordon and Betty Moore Foundation grant GBMF3834, and the Alfred P. Sloan Foundation grant 2013-10-27 to UC Berkeley through the Moore-Sloan Data Sciences Environment initiative at the Berkeley Institute for Data Science (BIDS). M.J.W was additionally supported by a diversity supplement of NHLBI R01HL117004, an Institutional Research and Academic Career Development Award K12GM081266, and an NHLBI Research Career Development (K) Award K01HL140218. M.G.C was additionally supported by NIH MARC U-STAR grant T34GM008574 at San Francisco State University. W.L.E was supported by grant R00HL135403. C.R. G was supported by grants R56HG010297 and T32HG00044. C.R.G was additionally supported by grants R56HG010297 and R01HG010297. The funders had no role in study design, data collection and analysis, decision to publish, or preparation of the manuscript.

**Competing interests:** We have read the journal's policy and one author of this manuscript (Chris Gignoux) has the following competing interests: ownership of stock in 23andMe. The remaining

## Author summary

Advances in RNA sequencing technology have reduced the cost of measuring gene expression at a genome-wide level. However, sequencing enough human RNA samples for adequately-powered disease association studies remains prohibitively costly. To this end, modern transcriptome-wide association analysis tools leverage existing paired genotype-expression datasets by creating models to predict gene expression using genotypes. These predictive models enable researchers to perform cost-effective association tests with gene expression in independently genotyped samples. However, most of these models use European reference data, and the extent to which gene expression prediction models work across populations is not fully resolved. We observe that these models predict gene expression worse than expected in a dataset of African-Americans when derived from European-descent individuals. Using simulations, we show that gene expression predictive model performance depends on both the proportion of genetic variants shared between population-specific prediction models as well as the genetic relatedness between populations. Our findings suggest a need to carefully select reference populations for prediction and point to a pressing need for more genetically diverse genotype-expression datasets.

## Introduction

In the last decade, large-scale genome-wide genotyping projects have enabled a revolution in our understanding of complex traits [1–4]. This explosion of genome sequencing data has spurred the development of new methods that integrate large genotype sets with additional molecular measurements such as gene expression. A recently popular integrative approach to genetic association analyses, known as a transcriptome-wide association study (TWAS) [5,6], leverages reference datasets such as the Genotype-Tissue Expression (GTEx) repository [7] or the Depression and Genes Network (DGN) [8] to link associated genetic variants with a molecular trait like gene expression. The general TWAS framework requires previously estimated *cis*-eQTLs for all genes in a dataset with both genotype and gene expression measurements. The resulting eQTL effect sizes build a predictive model that can impute gene expression in an independently genotyped population. A TWAS is similar in spirit to the widely-known genome-wide association study (GWAS) but suffers less of a multiple testing burden and can potentially detect more associations as a result [5,6].

Unlike a normal GWAS, where phenotypes are regressed onto genotypes, in TWAS the phenotype is regressed onto the imputed gene expression values, thus constituting a new gene-based association test. TWAS can also link phenotypes to variation in gene expression and provide researchers with additional biological and functional insights over those afforded by GWAS alone. While these models are imperfect predictors, predicted gene expression allows researchers to test phenotype associations to expression levels in existing GWAS datasets without measuring gene expression directly. In particular, these methods enable analysis of predicted gene expression in very large cohorts ($\sim 10^4$–$10^6$ individuals) rather than typical gene expression studies that measure expression directly ($\sim 10^2$–$10^3$ individuals). Several methods have been recently developed to perform TWAS in existing genotyped datasets. PrediXcan [6] uses eQTLs precomputed from paired genotype-expression data, such as those in GTEx, in conjunction with a new genotype set to predict gene expression. These gene expression prediction models are freely available online (PredictDB), creating resources for external researchers. Related TWAS approaches, such as FUSION[5], MetaXcan [9], or SMR [10], leverage eQTL

authors have declared that no competing interests exist.

information with GWAS summary statistics instead, thus circumventing the need for raw individual-level genotype data.

As evidenced by application to numerous disease domains, the TWAS framework is capable of uncovering new genic associations [11–17]. However, the power of TWAS is inherently limited by the data used for eQTL discovery. For example, since gene expression varies by tissue type, researchers must ensure that the prediction weights are estimated using RNA from a tissue related to their phenotype, whether that be the direct tissue of interest or one with sufficiently correlated gene expression [18]. Furthermore, the ability of predictive models to impute gene expression from genotypes is limited by the heritability in the *cis* region around the gene [6]. Consequently, genes with little or no measurable genetically regulated effect on their expression in the discovery data are poor candidates for TWAS.

A subtler but more troubling issue arises from the lack of genetic diversity present in the datasets used for predictive model training: most paired genotype-expression datasets consist almost entirely of data from European-descent individuals [8,18]. The European overrepresentation in genetic studies is well documented [19–21] and has severe negative consequences for equity as well as for gene discovery [22], fine mapping [23–25], and applications in personalized medicine [26–34]. Genetic architecture, linkage disequilibrium, and genotype frequencies can vary across populations, which presents a potential problem for the application of predictive models with genotype predictors across multiple populations.

The training data for most models in the models derived from PrediXcan weights in PredictDB (predictdb.org) are highly biased toward European ancestry: GTEx version v6p subjects are over 85% European, while the GTEx v7 and DGN subjects are entirely of European descent. The lack of suitable genotype-expression datasets in non-European individuals leads to scenarios in which PredictDB models trained in Europeans are used to predict into non-European or admixed populations. As shown previously in the context of polygenic risk scores [35], multi-SNP prediction models trained in one population can suffer from unpredictable bias and poor prediction accuracy that impair their cross-population generalizability. Recent analyses of genotype-expression data from the Multi-Ethnic Study of Atherosclerosis (MESA) [36–38], which includes non-European individuals, explore cross-population transcriptome prediction and conclude that predictive accuracy is highest when training and testing populations match in ancestry. These results are consistent with our experience analyzing admixed populations, but offer little insight into the mechanisms underlying the cross-population generalizability of transcriptome prediction models, particularly when eQTL architecture is known.

Here we investigate the cross-population generalizability of gene expression models using paired genotype and gene expression data and using simulations derived from real genotypic data and realistic models of gene expression. We analyze prediction quality from currently available PrediXcan prediction weights using a pilot subset of paired genotype and whole blood transcriptome data from the Study of African Americans, Asthma, Genes, and Environment (SAGE) [39–42]. SAGE is a pediatric cohort study of childhood-onset asthma and pulmonary phenotypes in African American subjects of 8 to 21 years of age. To tease apart cross-population prediction quality, we turn to GEUVADIS and the 1000 Genomes Project datasets, which includes multiple populations each with more samples than our SAGE cohort [4,43,44]. The GEUVADIS dataset has been used extensively to validate PrediXcan models [6,38]. However, recent analyses suggest that GTEx and DGN PrediXcan models behave differently on the constituent populations in GEUVADIS [45,46]. GEUVADIS provides us an opportunity to investigate predictive models with an experimentally homogeneous dataset: the GEUVADIS RNA-Seq data were produced in the same environment under the same protocol, from lymphoblastoid cell lines (LCLs) that, despite some variation in when cells were collected [47], are

derived from similar sampling efforts and treatments, thereby providing a high degree of technical harmonization. We train, test, and validate predictive models wholly within GEUVADIS with a nested cross-validation scheme. Finally, to understand the consequences of eQTL architecture on TWAS, we use existing 1000 Genomes data to simulate large samples of two ancestral populations and an admixed population and then apply the same "train-test-validate" scheme with various simulated eQTL models to study cross-population prediction efficacy when a gold standard is known.

## Results

### Concordance of measured gene expression and PrediXcan predictions is lower than expected

We compared transcriptome prediction accuracy in SAGE whole blood RNA using three PredictDB prediction weight sets for whole blood RNA: GTEx v6p, GTEx v7, and DGN. We also evaluated expression prediction with four of the MESA monocyte weight sets: MESA_ALL (populations combined), MESA_AFA (African Americans), MESA_AFHI (combined African Americans and Hispanic Americans), and MESA_CAU (Caucasians). For each gene where both measured RNA-Seq gene expression and predictions are available in SAGE, we compute both the coefficient of determination ($R^2$) and Spearman correlation to analyze the direction of prediction. As we are primarily interested in describing the relationship between predicted outcome and real outcome, we prefer Spearman's $\rho$ to describe correlations, while for determining prediction accuracy, we use the standard regression $R^2$, corresponding to the squared Pearson correlation, to facilitate comparisons to prior work. We then benchmark these against the out-of-sample $R^2$ and correlations from GTEx v7 and MESA as found in PredictDB. Prediction results in SAGE were available for 11,545 genes with a predictive model from at least one weight set. Not all sets derived models at the same genes: since the estimation of these prediction models requires both high quality expression data and inferred eQTLs, each weight set from PredictDB may have a different number of gene models. Therefore, intersecting seven different weight sets reduces the overall number of models available for comparison. After applying the recommended filters, the prediction results across all seven weight sets overlapped at 273 genes, of which 39 genes had predictions with positive correlation to measurements. These subsets contained genes that were expressed somewhat higher than average (S1 Fig). This small number of genes in common is largely driven by MESA_AFA, the repository with the smallest number of predictive models. However, MESA_AFA contains the models that should best reflect the genetic ancestry of African Americans in SAGE (S1 Table). We note that MESA_AFA also has the smallest training sample size among our weight sets (N = 233) [38], so the small number of predicted genes from MESA_AFA probably results from the small training sample size and not from any feature of the underlying MESA_AFA training data.

Here, we highlight the union of genes across model sets for investigation. The concordance between predicted and measured gene expression over the union of 11,545 from all seven weight sets, with corresponding training metrics from PredictDB as benchmarks, shows worse performance than expected for $R^2$ (Fig 1) and correlations (Fig 2). The highest mean $R^2$ of 0.0298 was observed in MESA_AFA, while the lowest was observed in MESA_ALL ($R^2$ = 0.0250), suggesting little appreciable difference in prediction quality between prediction weight sets. $R^2$ for SAGE cluster heavily around 0 (S2 Fig), while $R^2$ for the prediction weight repositories show a wider distribution of $R^2$ (S3 Fig). We note the intersection of all prediction models is limited, but reflects a similar pattern: results for the 273 common genes (S4–S6 Figs) and the 39 genes with positive correlations (S7 and S8 Figs) showed little difference from the

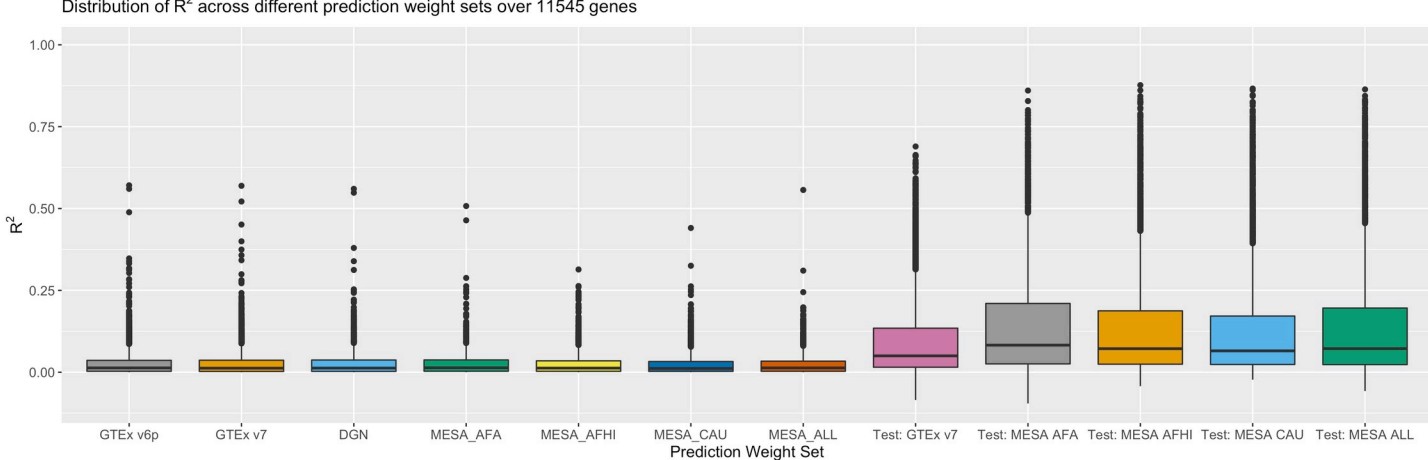

**Fig 1. A comparison of $R^2$ between prediction and measurement in SAGE, with PredictDB test metrics as benchmarks, for 11,545 genes total.** The prediction weights used here are, from left to right: GTEx v6p, GTEx v7, DGN, MESA African Americans, MESA African Americans and Hispanics, MESA Caucasians, and all MESA subjects. Test $R^2$ from model training in GTEx 7 and MESA ("test_R2_avg" in PredictDB) appear on the right and provide a performance baseline. The number of genes per weight set varies; see S1 Table.

$R^2$ shown in Fig 1. Because SAGE is an independent validation set for the training populations, we would expect to observe some deterioration in prediction $R^2$ due to out-of-sample estimation. However, Fig 1 shows a marked difference in model performance.

More noteworthy is the substantial proportion of predictions in SAGE with negative correlations to the real data. All seven weight sets produced gene expression predictions with negative correlations, but average performance across genes varied. The least negative mean correlation across prediction weight sets was observed in GTEx v6p (-0.0044), while the most negative mean correlation (-0.0204) was observed with MESA_AFA (MESA African Americans, S1 Table). The observation that correlations to SAGE measurements are sometimes negative on average suggests that some large $R^2$ values seen in Fig 1 may result from gene models with incorrect direction of prediction, thereby limiting interpretability of results. While there are some fluctuations in prediction accuracy, the fact that correlations vary from -0.0204 to

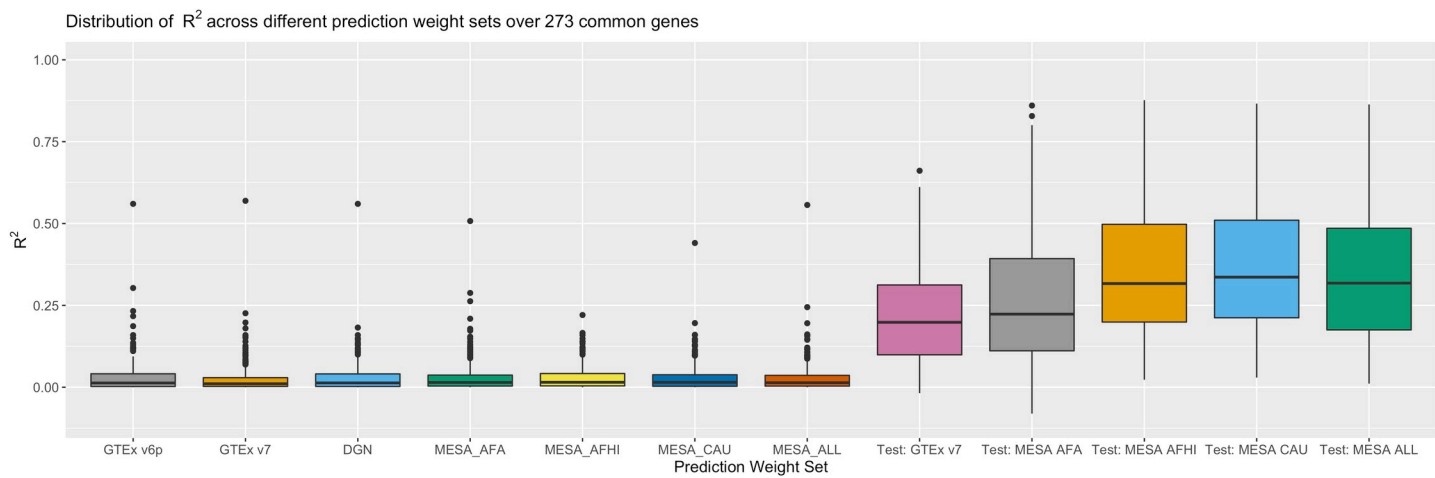

**Fig 2. Spearman correlations of measured gene expression versus predicted expression from PrediXcan.** The order of the weight sets matches Fig 1. Test correlations for GTEx v7 and MESA correspond to "rho_avg" from PredictDB.

-0.0044 indicates that no prediction weight set produces practically meaningfully better correlations to data than the others. In contrast, the published models for these genes show positive correlations to their training data, ranging from 0.308 in GTEx_v7 to 0.379 in MESA_AFA, indicating no obvious incapacity for accurate prediction, even with out-of-sample data. However, available predictions into SAGE from otherwise valid prediction models are uniformly limited in power to capture true genotype-expression relationships.

To analyze genes with high prediction $R^2$ in the original experiment, we focus on genes in GTEx v7 with cross-validated $R^2 > 0.2$ in the reference population. Our choice of $R^2$ is informed by observed $R^2$ between predictions and measurements in DGN (see Fig 3 of [6]) and focuses our analysis on genes predicted better than average. Fig 3 compares PredictDB testing $R^2$ against the empirical $R^2$ from regressing predictions onto observations in SAGE. In this case, even the better-imputed gene models derived from PredictDB have limited ability to capture gene expression accurately in SAGE (mean $R^2$ 0.031, IQR [0.0027, 0.037]). We see a similar trend with MESA models (S9–S12 Figs), in which $R^2$ in SAGE is consistently much lower (mean $R^2$ 0.026–0.030) than test $R^2$ from each prediction weight set (test $R^2$ 0.373–0.392).

Since SAGE data were ascertained on the basis of rs28450894 and by extension gene *NFKB1* [39], we checked if results were biased by ascertainment. Among the 273 genes in common to all weight sets, only one gene model, *SLC39A8*, lay within 1 megabase in either direction of rs28450894 on chromosome 4. Only two of the SNP predictors for *SLC39A8* showed more than moderate linkage disequilibrium ($R^2 > 0.2$) with rs28450894: SNP rs72696152 (MESA_ALL, $R^2 = 0.675$) and rs4648011 (DGN, $R^2 = 0.262$) (S3 Table). However, the resulting prediction quality were close to 0 like the remaining 272 genes, as the linear model $R^2$ for SLC39A8 ranged from 0.0007 (GTEx v7) to 0.0102 (GTEx v6p), indicating no obvious bias away from 0 (S4 Table).

## Cross-population prediction quality declines with increasing genetic distance

Real-world comparisons of RNA-Seq datasets can be subject to numerous sources of heterogeneity besides differential ancestry. Possible confounders include technical differences in sequencing protocols, differences in the age of participants [48] or cell lines [47], and the post-mortem interval to tissue collection (for GTEx) [49–51]. The small sample size of our SAGE cohort ($n = 39$) limits our ability to account for these possible confounders. To investigate cross-population generalizability in an experimentally homogeneous context, we turn to GEU-VADIS [43]. The GEUVADIS data include two continental population groups from the 1000 Genomes Project: the Europeans (EUR373), composed of 373 unrelated individuals from four subpopulations (Utahns (CEU), Finns (FIN), British (GBR), Toscani (TSI)), and the Africans (AFR) composed of 89 unrelated Yoruba (YRI) individuals. In light of the known bottleneck in Finnish population history [52], we analyze EUR373 both as one population and as two independent subgroups: the 95 Finnish individuals (FIN) and the 278 non-Finnish Europeans (EUR278). We used expression data, generated and harmonized together by the GEUVADIS Consortium, with matched whole-genome genotype data in the resulting four populations (EUR373, EUR278, FIN, and AFR) to train predictive models for gene expression in a nested cross-validation scheme [6] and perform cross-population tests of prediction accuracy.

Table 1 shows $R^2$ from three training sets (EUR373, EUR278 and AFR) into the four testing populations (EUR373, EUR278, FIN, and AFR) for genes with positive correlation between prediction and measurement. While the number of genes with applicable models including genetic data varies in each train-test scenario (see S5 Table), we note that not all predictive

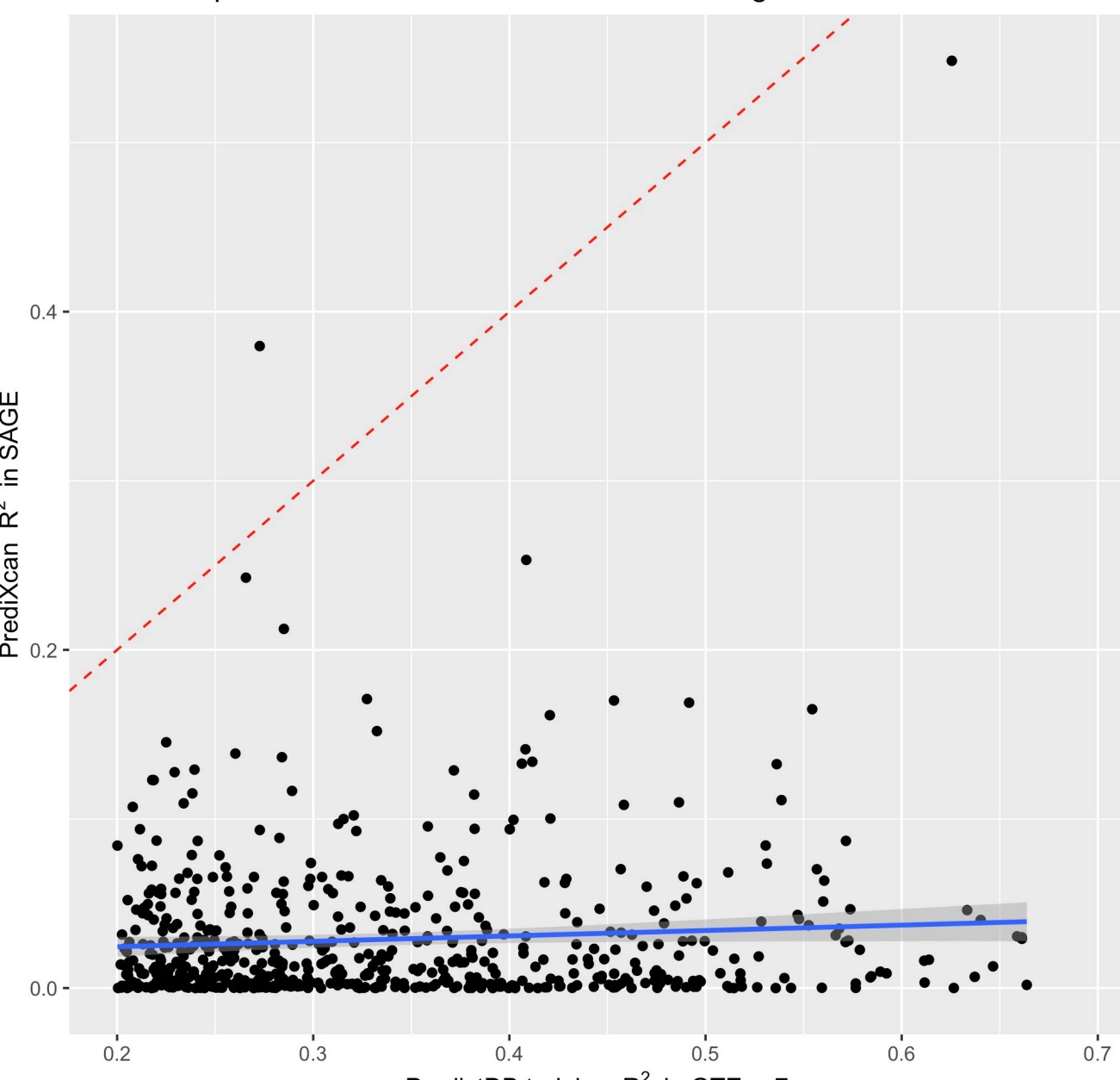

**Fig 3. A comparison of $R^2$ from SAGE and GTEx v7 training diagnostics.** The SAGE $R^2$ are computed from regressing PrediXcan predictions onto gene expression measurements. The GTEx v7 $R^2$ are taken from PredictDB ("test_R2_avg"). The red dotted line marks where $R^2$ between the two groups match, while the blue line denotes the best linear fit.

models are trained on equal sample sizes, so the resulting $R^2$ only provide a general idea of how well one population imputes into another; see S13 Fig for a distributional summary. Analyses within a population use out-of-sample prediction $R^2$ to avoid overfitting across train-test scenarios. Predicting from a population into itself yields $R^2$ ranging from 0.079–0.098 (Table 1) consistent with the smaller sample sizes in GEUVADIS versus GTEx and DGN. In contrast, predicting across populations yields more variable predictions, with $R^2$ ranging from

**Table 1. Prediction $R^2$ between populations in GEUVADIS for genes with positive correlation between predictions and measurements.** The number of genes analyzed in each scenario varied in each case; see S5 Table. Scenarios where the training sample is contained in the testing sample cannot be accurately tested and are marked with "n/a". EUR373 includes all 373 Europeans, EUR278 includes only the 278 non-Finnish Europeans, FIN includes only the 95 Finnish individuals, and AFR includes only the 89 Yoruba.

| $R^2$ | | Train Pop | | |
|---|---|---|---|---|
| | | EUR373 | EUR278 | AFR |
| Test Pop | EUR373 | 0.098 | n/a | 0.029 |
| | EUR278 | n/a | 0.096 | 0.030 |
| | FIN | n/a | 0.087 | 0.039 |
| | AFR | 0.054 | 0.051 | 0.079 |

0.029–0.087. At the lower range of $R^2$ (0.029–0.039) are predictions from AFR into European testing groups (EUR373, EUR278, and FIN). Alternatively, when predicting from European training groups into AFR, the $R^2$ are noticeably higher (0.051–0.054). Prediction from EUR278 into FIN ($R^2 = 0.087$) is better than prediction from EUR278 into AFR ($R^2 = 0.051$), suggesting that prediction $R^2$ may deteriorate with increased genetic distance. A comparison of the 564 genes in common across all train-test scenarios (Table 2) yields a subset of genes with potentially more consistent gene expression levels (see S14 Fig for distributional summaries). In this case involving better-predicted genes, we see that prediction quality between the European groups improves noticeably ($p$-value ~ 0, Dunn test). Among European training sets, the lowest $R^2$ is 0.183 for EUR278 predicting into EUR278. $R^2$ increases to 0.201 (EUR373 to EUR373) and attains its maximum at 0.216 (EUR278 to FIN), possibly a consequence of diminished haplotypic diversity from Finnish population bottlenecking as mentioned previously. In contrast, $R^2$ between Europeans and Africans ranges from 0.095 (AFR to EUR373) to 0.147 (EUR373 to AFR), a significant improvement ($p$-value $< 7.07 \times 10^{-22}$, Dunn test) that nonetheless highlights a continental gap in prediction performance. AFR predicts better into FIN ($R^2 = 0.111$) than the other European populations ($R^2 = 0.095$–0.096), similar to what we observe with predictions from EUR373 into FIN. But AFR predicts better into itself ($R^2 = 0.130$) than to other populations; similarly, European predictions into AFR are noticeably lower ($R^2 = 0.141$–0.147) than into other European populations ($R^2 = 0.183$–0.216). In general, populations seem to predict better into themselves, and less well into other populations.

Combining all European subpopulations obscures population structure and can complicate analysis of cross-population prediction performance. To that end, we divide the GEUVADIS data into its five constituent populations and randomly subsample each of them to the smallest population size ($n = 89$). We then estimate models from each subpopulation and predict into all five subpopulations. Table 3 shows average prediction $R^2$ from each population into itself and others (see S15 Fig for distributional summaries). The populations consistently predict

**Table 2. Prediction $R^2$ between populations in GEUVADIS for 564 gene models that show positive correlation between prediction and measurement in all 9 train-test scenarios that were analyzed.** Scenarios that were not tested are marked with "n/a". As before, EUR373 includes all 373 Europeans, EUR278 includes only the 278 non-Finnish Europeans, FIN includes only the 95 Finnish individuals, and AFR includes only the 89 Yoruba.

| $R^2$ (564 genes) | | Train Pop | | |
|---|---|---|---|---|
| | | EUR373 | EUR278 | AFR |
| Test Pop | EUR373 | 0.201 | n/a | 0.096 |
| | EUR278 | n/a | 0.183 | 0.095 |
| | FIN | n/a | 0.216 | 0.111 |
| | AFR | 0.147 | 0.141 | 0.130 |

**Table 3. Cross-population prediction performance across all five constituent GEUVADIS populations over genes with positive correlation between predictions and measurements.** All populations were subsampled to N = 89 individuals. The number of genes represented varies by training sample (CEU: N = 1029, FIN: N = 1320, GBR: 1436, TSI: 1250, YRI: 914).

| R2 Mean (Std Err) | | Training population | | | | |
|---|---|---|---|---|---|---|
| | | CEU | TSI | GBR | FIN | YRI |
| Testing Pop | CEU | 0.115 | 0.106 | 0.107 | 0.103 | 0.069 |
| | | (0.139) | (0.139) | (0.134) | (0.133) | (0.116) |
| | TSI | 0.124 | 0.121 | 0.124 | 0.118 | 0.083 |
| | | (0.158) | (0.151) | (0.149) | (0.145) | (0.13) |
| | GBR | 0.132 | 0.137 | 0.136 | 0.133 | 0.087 |
| | | (0.16) | (0.155) | (0.156) | (0.155) | (0.132) |
| | FIN | 0.128 | 0.130 | 0.130 | 0.130 | 0.084 |
| | | (0.158) | (0.155) | (0.153) | (0.152) | (0.134) |
| | YRI | 0.065 | 0.069 | 0.063 | 0.062 | 0.104 |
| | | (0.108) | (0.112) | (0.1) | (0.102) | (0.138) |

well into themselves, with prediction $R^2$ ranging from 0.104–0.136. We observe that prediction quality using models trained in CEU shows a miniscule decline relative to other EUR subpopulations. This observation is potentially due to the older age of CEU LCLs [45,53,54], but did not appreciably change our results. In contrast, a more notable difference exists between the EUR subpopulations and YRI. The cross-population $R^2$ between CEU, TSI, GBR, and FIN ranges from 0.103 to 0.137, while cross-population $R^2$ from these populations into YRI ranges from 0.062 to 0.084. Prediction between YRI and the EUR populations taken together is consistently lower than within the EUR populations (S17 Fig) and statistically significant (*p*-value $< 1.36 \times 10^{-4}$, Dunn test; see S8 Table). The cross-population differences remain for the 142 genes with positive correlation in all train-test scenarios (Table 4, S16 Fig), where $R^2$ for prediction into YRI ranges from 0.166 to 0.244, while $R^2$ within EUR populations ranges from 0.239 to 0.331. These results clearly suggest problems for prediction models that predict gene expression across populations, in similar regimes to those tested with linear predictive models and datasets of size consistent with current references. In addition, since AFR is genetically more distant from the EUR subpopulations than they are to each other, we interpret these results to imply that structure in populations can potentially exacerbate cross-population prediction quality (S18 Fig).

**Table 4. Cross-population prediction performance across all five subsampled GEUVADIS populations over the 142 genes with positive correlation between prediction and measurement in all 25 train-test scenarios.** As in Table 3, all populations were subsampled to n = 89 subjects.

| R2 Mean (Std Err) (142 genes, all positive correlation) | | Training population | | | | |
|---|---|---|---|---|---|---|
| | | CEU | TSI | GBR | FIN | YRI |
| Testing Pop | CEU | 0.239 | 0.269 | 0.291 | 0.297 | 0.201 |
| | | (0.18) | (0.177) | (0.166) | (0.168) | (0.164) |
| | TSI | 0.307 | 0.294 | 0.331 | 0.322 | 0.227 |
| | | (0.188) | (0.21) | (0.182) | (0.185) | (0.185) |
| | GBR | 0.320 | 0.326 | 0.318 | 0.350 | 0.235 |
| | | (0.175) | (0.181) | (0.191) | (0.178) | (0.183) |
| | FIN | 0.318 | 0.320 | 0.343 | 0.323 | 0.244 |
| | | (0.191) | (0.198) | (0.182) | (0.201) | (0.192) |
| | YRI | 0.166 | 0.205 | 0.195 | 0.189 | 0.213 |
| | | (0.164) | (0.163) | (0.157) | (0.156) | (0.177) |

## Admixture influences cross-population gene expression prediction quality under known eQTL architecture

The unresolved question is the extent to which these results hold with oracle knowledge of eQTL architecture, something impossible to investigate in real data when the causal links between eQTLs and gene expression can only be estimated. To investigate genomic architectures giving rise to gene expression, and in particular to investigate behavior in admixed populations, we forward-simulated haplotypes from HapMap3 [55] CEU and YRI using HAPGEN2 [56] and then sample haplotypes in proportions consistent with realistic admixture proportions (80% YRI, 20% CEU) [57] to construct a simulated African-American (AA) admixed population. We simulated $n = 1000$ samples for each population, a much larger sample than what is available in GEUVADIS and comparable to the training sample size of DGN or MESA_ALL PrediXcan models. We simulated eQTL architectures under an additive model of size $k$ causal alleles ($k = 1, 10, 20,$ and $40$) in the ancestral populations (CEU and YRI) and an expression phenotype with *cis*-heritability $h^2 = 0.15$ (recapitulating the average $h^2$ in DGN whole blood RNA-Seq data [6]) using the genomic background of genic regions on chromosome 22, thus testing various model sizes and LD patterns. To tease apart the effect of shared eQTL architecture, we allow the two ancestral populations CEU and YRI to share eQTLs with fixed effects in various proportions (0%, 10%, 20%, ..., 100%) to test a range of eQTL architectures. The admixed population AA always inherited all eQTLs from the two ancestral populations, which yielded different numbers of eQTLs per gene depending on how many eQTLs were shared by CEU and YRI. For example, for eQTL model size $k = 10$, when CEU and YRI shared all 10 eQTLs, then all three populations had the exact same 10 eQTLs. When CEU and YRI shared half of their eQTLs with each other, then each one had 5 population-specific eQTLs, and AA inherited 15 total eQTLs (5 unique to CEU, 5 unique to YRI, and 5 shared). If CEU and YRI shared no eQTLs, then all eQTLs were population-specific, and AA inherited 20 eQTLs (10 from CEU and 10 from YRI; see S19 Fig for an illustration). With these simulations providing known architectures for comparison, we then apply the train-test-validate scheme as before.

Fig 4 shows the cross-population Spearman correlations between predicted and simulated phenotypes in our simulated AA, CEU, and YRI, partitioned by proportion of shared eQTLs, for $k = 10$ causal eQTLs in the ancestral populations (CEU and YRI). Scenarios with k = 20 and k = 40 causal eQTLs show similar trends (S20 Fig and S21 Fig). Prediction within a population produced similar correlations in all cases, ranging from 0.310 to 0.338 (S6 Table). The case of models with 100% shared eQTL architecture–where eQTL positions and effects are exactly the same between the ancestral populations–yields predictions with no loss in cross-population generalizability, with correlations ranging from 0.299 to 0.336 even when predicting across populations (S7 Table). This case suggests that eQTLs that are causal in all populations can impute gene expression reliably regardless of the population in which they were ascertained, provided that the eQTLs can be correctly mapped and genotyped in all populations, that the eQTL effects are identical across populations, and that a linear model of eQTLs is assumed. For cases where eQTL architecture is not fully shared across populations, we see that prediction from each population into the other improves as the proportion of shared eQTLs increases (Fig 4). The cross-population correlation between predicted gene expression versus measurement is highest from YRI to AA (0.238 to 0.338), intermediate from CEU to AA (0.218 to 0.310), and lowest between CEU and YRI (0.0020 to 0.326). Prediction quality from AA to CEU and YRI interpolates that of YRI to AA and CEU to AA, with correlations ranging from 0.223 to 0.338. Prediction quality from AA to CEU or YRI shows a slight upward trend as more eQTLs are shared, an artifact of eQTL inheritance in our simulations; as

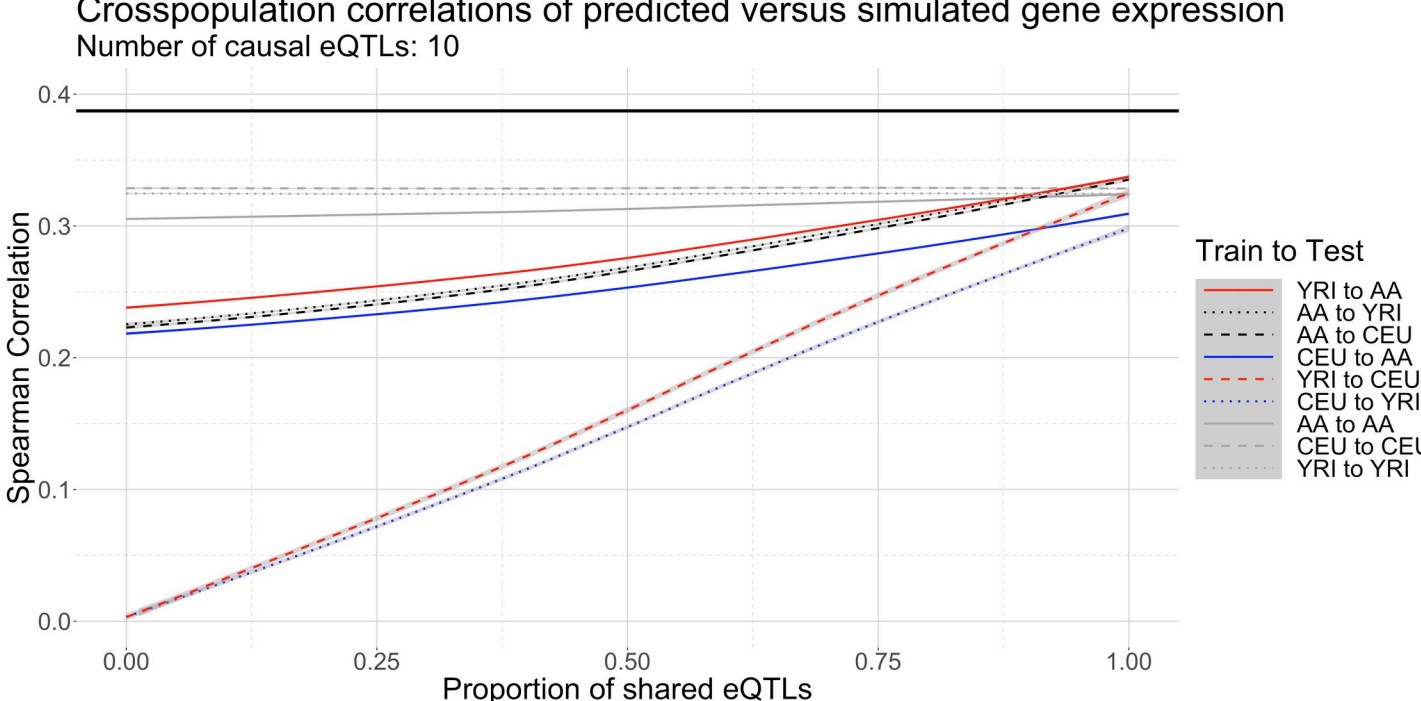

**Fig 4. Correlations between predictions and simulated gene expression measurements from simulated populations across various proportions of shared eQTL architecture with 10 causal cis-eQTLs.** Here YRI is simulated from the 1000 Genomes Yoruba, CEU is simulated from the Utahns, and AA is constructed from YRI and CEU. The black line represents the upper bound of correlation 0.387 dictated by our choice $h^2 = 0.15$ for the genetic heritability of expression. Each trend line represents an interpolation of correlation versus shared eQTL proportion. Gray areas denote 95% confidence regions of LOESS-smoothed mean correlations conditional on the proportion of shared eQTLs.

described previously, AA eQTL models are largest (20 eQTLs) when CEU and YRI share no eQTLs and smallest (10 eQTLs) when CEU and YRI share all eQTLs. Consequently, when predicting between two populations, the choice of which population is used to train predictive models can produce differences in prediction quality. Prediction quality between AA to CEU and AA to YRI is not significantly different (p-value ~ 1, Dunn test). All other train/test scenarios are significantly different from each other (S9 Table). The results for k = 10, 20, and 40 eQTLs show a consistent trend of prediction quality driven primarily by differences in eQTL architecture, with additional minor influence from ancestral similarity between populations (k = 10, Fig 4, similar plots in S20 Fig and S21 Fig). Although less realistic for most genes [5,6,18], we also analyzed models with a single causal eQTL. Trends for single-eQTL models are more difficult to analyze due the limitations of binary inference as to whether the causal SNP is identified or not. Nevertheless, when the causal eQTL is identified and shared across populations, prediction quality is high in all cases. If the causal eQTL differs across populations, then cross-population prediction between AA and YRI or CEU is noticeably better than prediction between CEU and YRI (S22 Fig), in line with results for other values of *k* that suggest that eQTL sharing is the primary driver of gene expression prediction quality.

## Power to detect associations declines with decreasing shared ancestry

Simulation of gene expression demonstrates that gene expression prediction quality is modulated by both shared eQTL architecture and shared genetic ancestry. These results suggest possible effects of cross-population generalizability on the power to detect associations between a

phenotype and gene expression measures in a TWAS. For each of our three populations (AA, CEU, and YRI), we used the simulated gene expression measures to simulate a continuous phenotype whose variation depends on expression of a single causal gene. For simplicity, the phenotypes shared the same causal gene, the same effect size, and the same environmental noise model. We tested various effect sizes from $1 \times 10^{-5}$ to 1 and drew the environmental noise from a zero-mean normal distribution with variance 0.01. The effect sizes produced a continuous spectrum of genetic heritability values $h^2$ spanning the full range of heritability for gene expression. We then regressed the phenotype onto predicted gene expression measures, resulting in nine association tests, one for each train-test scenario. For simplicity, we focused on the prediction scenario with $k = 10$ causal eQTLs per gene. To see how shared eQTL architecture affects power, we used predicted expression measures with 0%, 50%, and 100% shared eQTLs per gene.

Fig 5 shows power curves for the association tests for the nine prediction scenarios for all three tested eQTL architectures. Unsurprisingly, power improves as populations share more causal eQTLs and as more phenotypic heritability is driven by gene expression. For example, with 100% shared eQTLs and phenotypic heritability 0.205, cross-population power ranges between 0.69–0.86. In contrast, average power under a model of 0% shared eQTLs between ancestral populations (CEU to YRI, YRI to CEU), varies from a scant 0.02 to 0.14 as phenotype heritability increases from 0 to 1, indicating some ability to predict gene expression at genetically controlled genes even without shared eQTLs. AA shows better cross-population power, ranging from to 0.38 (AA to YRI, AA to CEU) to 0.82 (CEU to AA) and 0.88 (YRI to AA), an expected outcome since AA inherits all eQTLs from the ancestral populations. Power also improves with shorter genetic distance between populations. Fig 6, which is a cross-section of Fig 5, shows power for each train-test scenario across various shared eQTL architectures for $\beta = 0.05$, corresponding to a phenotype heritability of $h^2 = 0.205$, indicating moderate genetic control. TWAS in this case using gene expression imputed from matched populations has higher power across all eQTL architectures, from 0.88–1.00, compared to cross-population TWAS, where power varies substantially. For an architecture with no shared eQTLs, power between CEU and YRI is 0.02, while power is higher for CEU to AA (0.82) and YRI to AA

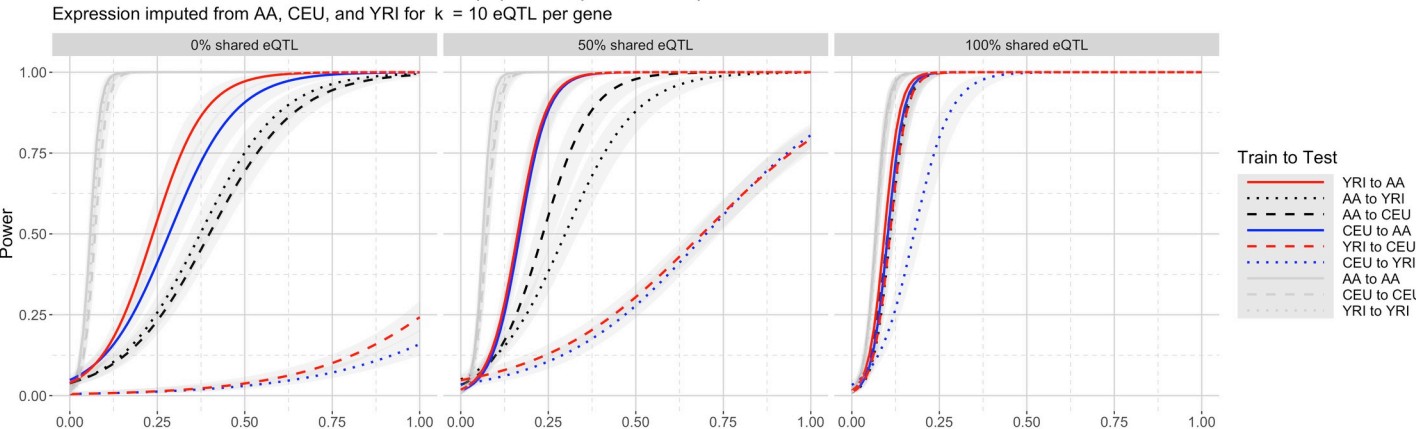

**Fig 5. Curves depicting power to detect association under various TWAS scenarios.** The x-axis represents the proportion of phenotypic variance explained by gene expression. As in *Fig 4*, AA reflects simulated African-Americans constructed from YRI and CEU. The curves represent logistic interpolations of whether or not the causal gene was declared significant in an association test of a phenotype from the testing population with gene expression predicted from a training population into the testing population. Gray areas denote 95% confidence regions of mean power conditional on the effect size.

(0.86). TWAS power for expression predicted from AA to CEU (0.39) or YRI (0.38) is much lower due to the aforementioned structure of eQTL inheritance; when CEU and YRI share 0 of 10 eQTLs, AA has 20 eQTLs, 10 from each ancestral population. As the proportion of shared eQTLs jumps from 0% to 50% and 100%, power increases across all cross-population scenarios, reaching up to 0.86 (YRI to AA, 100% shared eQTLs). When eQTLs are fully shared, power from YRI to AA (0.86) is higher than from CEU to AA (0.83), indicating an effect of genetic distance on prediction quality. Indeed, when controlling for eQTL architecture, increasing genetic similarity between reference and target populations yields more significant median association test *t*-statistics (S23 Fig).

## Admixture proportion interpolates power in two-way admixture

The results in Fig 6 show how genetic distance affects power in TWAS association tests for one particular admixture proportion, but offer limited insight about how power changes across the admixture spectrum. To understand how admixture proportion affects TWAS power in a general admixed population with two ancestral populations, we simulated multiple admixed populations from CEU and YRI with admixture proportions varying at 10% increments. When the admixed population has 0% YRI admixture, it is fully drawn from haplotypes from CEU, whereas a population with 100% YRI admixture is drawn exclusively from haplotypes from YRI. It is important to note that in neither case does the admixed population exactly match the reference CEU or YRI since the genotypes for the admixed population are formed from an independent shuffling of the CEU or YRI haplotypes. For each admixed population, we estimated prediction models of gene expression as done in our previous analyses. For computational efficiency, we investigated the scenario of 50% shared eQTLs across reference populations and the number of eQTLs per gene equal to 10. Populations still shared the same causal gene, effect size, and environmental noise model.

Fig 7 shows power across admixture proportions for all cross-population scenarios. The phenotypes were simulated at effect sizes $\beta = 0.005$, 0.01, and 0.025, and environmental variance $\sigma^2 = 0.01$, corresponding to heritability $h^2 = 0.06$, 0.20, and 0.58, respectively. To compare and contrast across each train and test scenario, we plot the overall trends of performance in

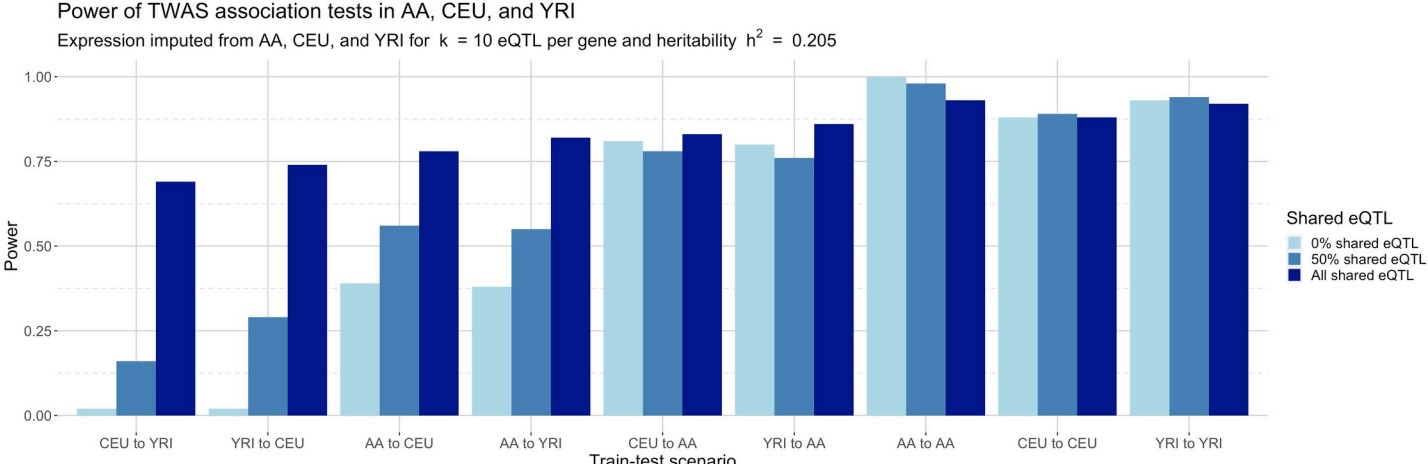

**Fig 6. Power for phenotype-expression association tests with cross-population imputed gene expression for heritability $h^2 = 0.205$.** The cross-population scenarios are ordered left to right from least admixture (CEU to YRI, 0% admixture proportion in our simulation) to most admixture (YRI to AA, 80% admixture proportion). Power increases on two axes: (1) as the proportion of shared eQTL architecture increases, and, to a lesser extent, (2) as genetic distance decreases between reference and target populations. Power is consistently high when training and testing populations match.

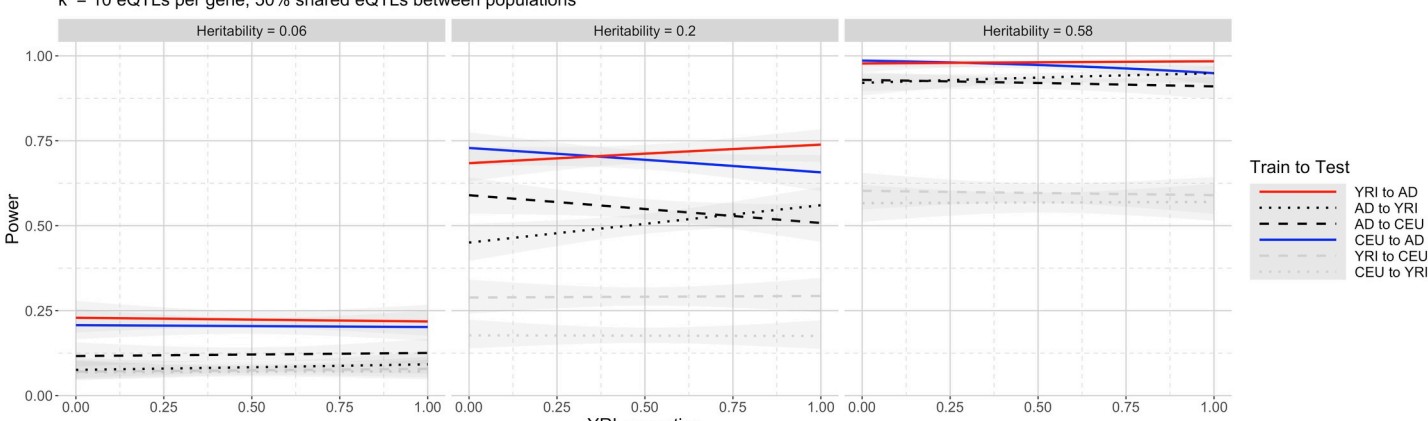

**Fig 7. Power for various cross-population train-test scenarios with varying YRI admixture for three phenotypic heritability levels h² = 0.06, 0.20, and 0.58, corresponding to effect sizes 0.005, 0.01, and 0.025, respectively.** Power increases as heritability increases, but also as populations become more genetically similar. Raw power estimates and 95% confidence intervals are listed in S10–S12 Tables.

Fig 7, and provide the exact mean power estimates and 95% confidence intervals for each scenario in S10–S12 Tables. To avoid confusion with previous references to AA, which had a fixed admixture proportion, here we denote the admixed population for all proportions as AD. As expected, statistical power increases with the genetic heritability of the phenotype for all prediction scenarios. However, the different admixture proportions yield directional changes in power when gene expression is predicted to or from AD. For example, when $h^2 = 0.20$ and gene expression is predicted from AD to CEU, power at 0% YRI admixture is 0.56 (95% CI: 0.462–0.658) and declines linearly with increasing YRI admixture; at 100% YRI, statistical power for AD to CEU is 0.46 (95% CI: 0.362–0.558). For AD to YRI, power at 0% YRI admixture starts at 0.42 (95% CI: 0.323–0.512) and increases linearly to 0.53 (95% CI: 0.431–0.628) at 100% YRI. We observe similar changes in power for CEU to AD (decreasing power as YRI proportion increases) and YRI to AD (increasing power as YRI proportion increases). The four directional trends also hold for $h^2 = 0.06$ and $h^2 = 0.58$, though power for cross-population scenarios involving AD is much lower in the former case and almost universally high in the latter case. In essence, the varying admixture proportions in this two-way admixed population yield a continuous linear trend of statistical power between the two ancestral populations: when AD is genetically closer to CEU, power for gene expression predicted in these populations is highest, and declines as AD becomes genetically closer to YRI. Similarly, when predicting from AD to YRI or vice versa, power is lowest when the two populations are genetically distinct, intermediate as the two populations become more genetically similar, and maximized when they are most alike.

## Discussion

Our goal with this study was to understand the extent to which gene expression prediction models estimated in one population can accurately predict the genetic component of gene expression in a different population. Cross-population generalizability of gene expression prediction models is an important but understudied issue for TWAS analyses. Among TWAS resources, we focused on PrediXcan as a test case with openly distributed prediction models available for multiple populations [6,38]. Using 39 subjects from the SAGE study [39–42] we compared predicted expression values from PrediXcan models to measured gene expression

on the same subjects and found that predictions matched poorly to measurements. Our investigation with the GEUVADIS dataset [43] offered us a more homogenous environment and larger sample size in which to train and test gene expression prediction models. Prediction quality in GEUVADIS using both continental and constituent subpopulations provided stronger evidence of cross-population generalizability issues with transcriptome prediction, but could not control for eQTL predictors that vary between populations. To that end, our simulation of an admixed population from 1000 Genomes CEU and YRI haplotypes [4,44] allowed us to finely control eQTL positions and effects as well as the causal genes in a TWAS. The simulation results show that both gene expression prediction accuracy and statistical power decrease as population eQTL models begin to diverge and genetic distance increases between populations for varying admixture proportions.

Our results highlight two points: firstly, since prediction within populations is better than prediction between populations, our results reaffirm prior investigations [38] that population matching matters for optimally predicting gene expression. This is consistent with our results of impaired transcriptome prediction performance in SAGE with currently available resources. Secondly, despite decreased prediction accuracy when predicting between different populations, the populations that are more closely genetically related demonstrate somewhat better cross-population prediction and power to detect associations in TWAS. Our simulations of prediction between ancestral populations and an admixed one under varying admixture proportions neatly summarize this relationship: the admixture proportion from each ancestral population interpolates the power available from each ancestral population, and power is maximized when the admixed population is most closely related to one or the other ancestral population. However, while the differences in power under varying admixture are statistically meaningful, they are smaller than differences attributable to different eQTL architectures or to different levels of genetic heritability of a phenotype.

Prediction results from GTEx, DGN, and MESA into SAGE suggest that current predictive models, even for genes with greater heritability, perform worse than expected despite matching tissue types. Our investigation into cross-population prediction accuracy with GEUVADIS data replicates this lack of cross-population generalizability as observed with current predictive models from PredictDB, demonstrating that heterogeneity in RNA-Seq protocols does not fully explain our observations. Our results parallel prior evidence [45] that PredictDB models themselves do not predict as well as expected into GEUVADIS despite controlling for tissue type, strongly suggesting that our observations about PrediXcan predictions in SAGE could hold true in other datasets. Since transcriptome prediction models use multivariate genotype predictors trained on a specific outcome, the impaired cross-population application can be viewed as an analogous observation to that seen previously in polygenic scores [35].

Our simulations control for many technical issues that are otherwise difficult to overcome with real data, such as oracular knowledge of positions and effect sizes of causal eQTLs. Nevertheless, in our simulations we see issues with cross-population prediction that we first observed when applying existing PrediXcan models to SAGE genotype data. Certainly, SAGE differs in important ways from GTEx, DGN, and MESA: SAGE is a pediatric asthma case-control cohort study in African-American children, so we cannot rule out technical heterogeneity introduced by differences in age, study design, and ethnicity. Furthermore, our SAGE sample includes RNA-Seq data for $n = 39$ subjects, a dataset leveraged previously to validate genetic associations, but is nevertheless somewhat small by contemporary standards [39]. However, technical heterogeneity between SAGE and existing PrediXcan models cannot solely explain the poor prediction performance. Our simulation results strongly suggest that problematic cross-population prediction performance between PrediXcan models and SAGE is deeper than differences in expression data.

Our investigations into the architecture of gene expression indicate that the power to detect associations is primarily determined by the degree of shared eQTLs across populations. In our simulations, this can be approximated as a (quasi-)linear interpolation of the prediction in the ancestral or reference populations into the admixed populations. However, the same is not true of overall levels of power in the admixed population: under 100% shared eQTL scenarios, cross-population generalizability is high, so the choice of training population matters less. In practical terms, this result bodes well for prediction of genes with eQTLs that do not vary by population. It is curious that in high-heritability genes, even models that share no eQTLs still retain power to detect scenarios: for genetically distant populations (CEU and YRI), power ranges from 0.10–0.14. Without shared eQTLs, this implies that local linkage disequilibrium between population-specific eQTLs, combined with high heritability, enables some degree of cross-population prediction. When cross-population statistical power is driven by LD and $h^2$ instead of expression signals, then subsequent interpretation of association hits, such as direction and strength of effect, becomes difficult to link to actual biological relationships between phenotype and gene expression.

It is important to note that our observations do not reflect shortcomings of either the initial PrediXcan or TWAS frameworks. Nor do our findings affect the positive discoveries made using these frameworks over the past several years. These methods fully rely on the data used as input for training, and the most commonly used datasets for model training are overwhelmingly of European descent. Here we note that the current models fail to capture the complexity of the cross-population genomic architecture of gene expression for populations of non-European descent. Failing to account for this could lead researchers to draw incorrect conclusions from their genetic data, particularly as these models would lead to false negatives.

To this end, our simulations strongly suggest that predicting gene expression in a target population is improved by using predictive models constructed in a genetically similar training population. Maximizing prediction quality crucially depends on both genetic architecture and eQTL architecture. If populations share the exact same eQTL architecture, then they are essentially interchangeable for the purposes of gene expression prediction so long as eQTLs are genotyped and accurately estimated, which remains a technological and statistical challenge. As the proportion of shared eQTL architecture decreases between two populations, both cross-population prediction quality and TWAS power decrease as well. In both SAGE and GEUVADIS, we observe cross-population patterns consistent with an imperfect overlap of eQTLs across populations. Ensuring representative eQTL architecture for all populations in genotype-expression repositories will require a solid understanding of true cross-population and population-specific eQTLs. However, expanding the amount of global genetic architecture represented in genotype-expression repositories, which can be accomplished by sampling more populations, provides the most desirable course for improving gene expression prediction models. Additionally, this presents an opportunity for future research in methods that could improve cross-population generalizability, particularly when one population is over-represented in reference data. Tools from transfer learning could facilitate porting TWAS eQTL models from reference populations to target populations using little or no RNA-Seq data.

In light of the surging interest in gene expression prediction and TWAS, we see a pressing need for freely distributed predictive models of gene expression estimated from coupled transcriptome-genome data sampled in a variety of populations and tissues. The recently published predictive models with multi-ethnic MESA data constitute a crucial first step in this direction for researchers working with admixed populations [38]. However, the clinical and biomedical research communities must push for more diverse genotype-expression resources to ensure that the fruits of genomic studies benefit all populations.

### Online resources

PredictDB: http://predictdb.org/
GTEx: http://gtexportal.org/
DGN: http://dags.stanford.edu/dgn/
GEUVADIS: https://www.ebi.ac.uk/Tools/geuvadis-das/
Source code: https://github.com/asthmacollaboratory/sage-geuvadis-predixcan
Results and simulation data: https://doi.org/10.7272/Q6RN362Z

## Methods

### Ethics statement

This study uses data from the Study of African Americans, Asthma, Genes, and Environments (SAGE) cohort, approved for human subjects research under expedited review by UCSF IRB 10–02877 with reference #244919. All subjects gave written consent for genotyping, phenotyping, and data usage for general research use.

### Genotype and RNA-Seq data

RNA-Seq (RNA sequencing) data generation and cleaning protocols for 39 SAGE subjects analyzed here were initially described in (Mak, White, Eckalbar, et al. 2018) [39]. Genotypes were generated on the Affymetrix Axiom array as described previously [58]. Genotypes were then imputed on the Michigan Imputation Server [59] with EAGLE v2.3 [60] and the 1000 Genomes panel phase 3 v5 [44] and then subjected to the following filters: missing samples < 5%, missing genotypes at any given SNP <5%, SNP minor allele frequency > 1%, p-value for deviation from Hardy-Weinberg Equilibrium >1 x $10^{-4}$, and genotype imputation $R^2 > 0.3$. The choice of the 1000 Genomes panel follows GTEx protocol, though GTEx used the smaller 1000 Genomes phase 1 panel [4]. Gene expression counts were processed through the GTEx v6p eQTL quality control pipeline and as described previously [18]. Per GTEx protocol, gene expression values were corrected for 3 genotype principal components, 15 PEER factors, and sex. Gene expression values were filtered to have >0.1 reads per kilobase per million reads (RPKM) in at least 10 individuals and at least 6 reads in at least 10 individuals. This filtering process kept 20,985 genes with Ensembl identifiers for analysis, of which 20,268 were autosomal genes. We then quantile normalized the remaining gene expression values across samples as our gene expression measurements.

GEUVADIS genotype VCF files and normalized gene expression data (filename GD462. GeneQuantRPKM.50FN.samplename.resk10.txt.gz) were downloaded directly from the EMBL-EBI GEUVADIS Data Browser. Genotypes were filtered similarly to SAGE subjects. No manipulation was performed on expression data. This process yielded 23,722 genes for analysis.

### Running PrediXcan models

We ran PrediXcan on SAGE subjects using PredictDB prediction weights from three paired genotype-expression datasets from PredictDB: GTEx, DGN, and MESA [6,9,38,61]. For GTEx, we used both GTEx v6p and GTEx v7 weights. For MESA, we used all weight sets from the freeze dated 2018-05-30: African Americans (MESA_AFA), African Americans and Hispanics (MESA_AFHI), Caucasians (MESA_CAU), and all MESA samples (MESA_ALL). Overall, the analysis included 11,545 genes, of which only 273 had *both* normalized RNA-Seq measures and predictions from *all* weight sets. Of these, 126 had positive correlation between prediction and measurement. We assessed prediction quality by comparing PrediXcan predictions to normalized gene expression from SAGE using linear regression and correlation tests.

## Estimation of prediction models

We trained prediction models in GEUVADIS on genotypes in a 500Kb window around each of 23,723 genes with measured and normalized gene expression. GEUVADIS subjects were partitioned into various groups: the Europeans (EUR373), the non-Finnish Europeans (EUR278), the Yoruba (AFR), and the constituent 1000 Genomes populations (CEU, GBR, TSI, FIN, and YRI). For each training set, we performed nested cross-validation. The external cross-validation for all populations used leave-one-out cross-validation (LOOCV). The internal cross-validation used 10-fold cross-validation for EUR373 and EUR278 and LOOCV for the five constituent GEUVADIS populations in order to fully utilize the smaller sample size ($n$ = 89) compared to EUR278 ($n$ = 278) and EUR373 ($n$ = 373). Internal cross-validation used elastic net regression with mixing parameter $\alpha$ = 0.5 as implemented in the glmnet package in R. The nonzero weights for each SNP from each LOOCV were compiled and averaged for each gene, yielding a single set of prediction weights for each gene. Predictions were computed by parsing genotype dosages from the target population corresponding to the nonzero SNP predictors, and then multiplying dosages against the prediction weights. The resulting predictions were compared to normalized gene expression measurements downloaded from the GEUVADIS data portal. We applied two additional filters to ensure that gene expression models were suitable for analyses. Firstly, we removed genes that did not have any eQTLs in their predictive models. Secondly, genes where fewer than half of the individuals had nonmissing predictions were removed from further analysis. This latter filter discarded those genes for which expression was not easy to predict across multiple samples. Coefficients of determination ($R^2$) were computed with the lm function in R. Spearman correlations were computed with the cor.test function in R.

## Simulation of gene expression

We downloaded a sample of 20,085 HapMap 3 SNPs [55] from each of CEU and YRI on chromosome 22 as provided by HAPGEN2 [56]. The data include 234 phased haplotypes for CEU and 230 phased haplotypes for YRI. We forward-simulated from these haplotypes to obtain two populations of $n$ = 1000 individuals each. We then sampled haplotypes in proportions of 80% YRI and 20% CEU to obtain a mixture of CEU and YRI where the ancestry patterns roughly mimic those of African Americans. For computational simplicity, and in keeping with the high ancestry LD present in African Americans [62,63], for each gene we assumed local ancestry was constant for each haplotype. For each of the three simulated populations, we applied the same train-test-validate scheme used for cross-population analysis in GEUVADIS. Genetic data for model simulation were downloaded from Ensembl 89 and included the largest 100 genes from chromosome 22. We defined each gene as the start and end positions corresponding to the canonical transcript, plus 1 megabase in each direction. We removed two genes, *PPP6R2* and *MOV10L1*, that spanned no polymorphic markers within 2 megabases of their start and end positions in the HapMap3 dataset, resulting in 98 gene models used for analysis. To simulate predictive eQTL models, we tested multiple parameter configurations for each gene: we varied the number of causal eQTLs in the ancestral populations ($k$ = 1, 10, 20, and 40) and the proportion of shared eQTL positions ($p$ = 0.0, 0.1, 0.2, . . ., 0.9, 1) between ancestral populations. The admixed population always inherited all eQTLs from the ancestral populations. Causal eQTLs were chosen at random among SNPs with at least 5% minor allele frequency. The same 5% minor allele frequency floor was applied to each population. Each model included a simulated gene expression phenotype with *cis*-heritability set to 0.15. For each parameter configuration, we ran 100 different random instantiations of the model simulations.

## Simulation of TWAS

Using the simulated gene expression measures with $k = 10$ causal eQTLs per gene in ancestral populations, we simulated a continuous phenotype with a known genetic architecture that depended on 1 causal gene. We tested prediction scenarios with 0%, 50%, and 100% eQTLs shared across populations. For each eQTL architecture, the three populations AA, CEU, and YRI shared the same causal gene $G$, the same causal effect size $\beta$, and the same environmental noise $\varepsilon$. $G$ was chosen randomly. Effect sizes were fixed, and we tested various effect magnitudes $\beta = 1 \times 10^{-5}, 5 \times 10^{-5}, 1 \times 10^{-4}, \ldots, 1 \times 10^{-1}, 5 \times 10^{-1}, 1$, yielding a spectrum of phenotype heritability explained by gene expression. The environmental noise $\varepsilon$ was drawn from an $N(0, 0.1^2)$ distribution. Consequently, phenotypes therefore only varied with the expression measures from $G$. For a given population $c$, the phenotype $y_c$ was then simulated as

$$y_c = G\beta + \varepsilon.$$

For each combination of shared eQTL architecture, $G$, and $\beta$, this procedure yielded one $y_c$ per individual in a population. We then performed a TWAS with $y_c$ onto the *predicted* gene expression values, yielding three TWAS per $y_c$, one for each reference prediction population. We then queried the resulting association $p$-value at G and tabulated whether it was declared significant (yes) or not (no) against a Bonferroni-corrected threshold of 0.05 / 98, accounting for all 98 genes in the TWAS. We ran this procedure for 100 random instantiations of $(G, \varepsilon)$ and computed association test power with a logistic interpolation of the yes/no results.

## Analysis tools

Analyses used GNU parallel [64]. The R packages used for analysis include argparser, assertthat, data.table, doParallel, dunn.test, knitr, optparse, peer, the Bioconductor packages annotate, biomaRt, and preprocessCore, and the tidyverse bundle [65–76]. All plots were generated with ggplot2 [77].

## Supporting information

**S1 Table. Summary statistics for analyzing gene expression prediction in SAGE for all seven weight sets in PredictDB.** SAGE has measurements for 20,985 genes, of which 20,268 are autosomal. The intersection of genes with both predictions and measurements in SAGE across all seven weight sets is 273 (see S4–S6 Figs), of which 39 produce predictions positively correlated to data in all comparisons (see S7 and S8 Figs).
(XLSX)

**S2 Table. Summary statistics for each filtering step in the analysis of gene expression models from GEUVADIS for the 3 training populations EUR373, EUR278, and AFR.** The analysis of prediction vs. measurement contains 5038 genes in common between all three populations. Of these genes, 1476 genes demonstrate positive correlation between predictions and measurements.
(XLSX)

**S3 Table. SNPs in linkage disequilibrium with rs28450894, on which SAGE RNA-Seq data were ascertained.** Each SNP is a prediction weight in at least one prediction weight set (Prediction Weights). SNPs corresponding to the largest two $R^2$ values for each repository are listed here.
(XLSX)

**S4 Table. R2 between predictions and observations for gene SLC39A8 (Ensembl ID ENSG00000138821).** Some predictors for *SLC39A8* are in linkage disequilibrium with SNP rs28450894, but the resulting $R^2$ are not obviously biased away from 0.
(XLSX)

**S5 Table. Summary statistics from training and testing results with continental GEUVA-DIS populations for gene models with positive correlations.** The $R^2$ correspond to Table 1. The column "Correlation" lists the Spearman correlations for each scenario, while "Transcripts" gives the number of gene models used to compute the $R^2$ and correlation summaries.
(XLSX)

**S6 Table. Spearman correlations between prediction versus simulated measurement from simulated populations to themselves across various shared eQTL proportions for k = 10 causal eQTLs.**
(XLSX)

**S7 Table. Prediction performance under fully shared eQTL architecture for k = 10 eQTLs yields reliable cross-population gene expression prediction.** Results for other sizes of eQTL models are similar.
(XLSX)

**S8 Table. A Dunn test shows statistically significant differences when predicting between AFR and EUR populations versus predicting between EUR populations.**
(XLSX)

**S9 Table. Differences in cross-population prediction performance are statistically significant, with a few notable exceptions.** Prediction from AA to CEU or YRI is essentially the same, but all other scenarios are different, indicating that the direction of prediction does matter.
(XLSX)

**S10 Table. Power estimates and 95% confidence intervals for each train-test scenario (Train-Test) and each proportion of YRI (YRI proportion) corresponding to the left panel of Fig 7 for effect size 0.005 ($h^2$ = 0.06).**
(XLSX)

**S11 Table. Power estimates and 95% confidence intervals for each train-test scenario (Train-Test) and each proportion of YRI (YRI proportion) corresponding to the center panel of Fig 7 for effect size 0.01 ($h^2$ = 0.20).**
(XLSX)

**S12 Table. Power estimates and 95% confidence intervals for each train-test scenario (Train-Test) and each proportion of YRI (YRI proportion) corresponding to the right panel of Fig 7 for effect size 0.01 ($h^2$ = 0.58).**
(DOCX)

**S1 Fig. Distributions of log-transformed values of transcripts-per-million (TPM) from GTEx v7.** The overall distribution of gene expression values from GTEx v7 ("All GTEx v7 genes") shows lower average gene expression than the 273 genes in common across all PrediXcan sets ("Common genes") as well as the 39 of those genes with positive correlation. TPM values were downloaded directly from GTEx v7 (gtexportal.org).
(TIFF)

**S2 Fig. Distribution of $R^2$ in SAGE.** In contrast to Fig 1, the comparison to test $R^2$ from PredictDB is removed to facilitate comparison of prediction weight sets in SAGE. The weight sets

are ordered from left to right: GTEx v6p, GTEx v7, DGN, MESA African Americans, MESA African Americans and Hispanics, MESA Caucasians, and all MESA subjects.
(TIFF)

**S3 Fig. Violin plots of $R^2$ between predictions and measurements in SAGE, with testing $R^2$ from each PrediXcan repository included for benchmarking.** The prediction weights used here are, from left to right: GTEx v6p, GTEx v7, DGN, MESA African Americans, MESA African Americans and Hispanics, MESA Caucasians, and all MESA subjects. Test $R^2$ from model training in GTEx 7 and MESA ("test_R2_avg" in PredictDB) appear on the right and provide a performance baseline. $R^2$ for SAGE cluster heavily near zero, while testing $R^2$ from each repository are more evenly distributed.
(TIFF)

**S4 Fig. $R^2$ of measured gene expression versus predictions from PrediXcan.** The prediction weights used here are, from left to right: GTEx v6p, GTEx v7, DGN, MESA African Americans, MESA African Americans and Hispanics, MESA Caucasians, and all MESA subjects. Test $R^2$ from model training in GTEx 7 and MESA ("test_R2_avg" in PredictDB) appear on the right and provide a performance baseline.
(TIFF)

**S5 Fig. A violin plot of the $R^2$ as shown in S4 Fig.** Compared to the distributions from S3 Fig, the test $R^2$ from PrediXcan repositories show that these 273 genes are somewhat better predicted on average compared to all 11,545 genes shown in Fig 1. However, $R^2$ in SAGE are still heavily biased towards 0, indicating no obvious change in prediction quality.
(TIFF)

**S6 Fig. $R^2$ in SAGE for all PrediXcan prediction weight sets, similar to S4 Fig, but without testing $R^2$ from PredictDB.** The distributions are taken over the 273 genes in common to all weight sets.
(TIFF)

**S7 Fig. $R^2$ between prediction and measurement in SAGE only using the 39 genes with positive correlation between prediction and measurement in all weight sets and benchmarks.**
(TIFF)

**S8 Fig. $R^2$ in SAGE for all PrediXcan prediction weight sets, similar to S7 Fig, but without testing $R^2$ from PredictDB.** The distributions are taken over the 39 genes with positive correlation in all weight sets.
(TIFF)

**S9 Fig. A comparison of $R^2$ from SAGE and MESA_ALL training diagnostics, similar to Fig 3.** The SAGE $R^2$ are computed from regressing PrediXcan predictions onto gene expression measurements. The MESA_ALL $R^2$ are taken from PredictDB ("test_R2_avg").
(TIFF)

**S10 Fig. A comparison of $R^2$ from SAGE and MESA_AFA training diagnostics, similar to Fig 3.** The SAGE $R^2$ are computed from regressing PrediXcan predictions onto gene expression measurements. The MESA_AFA $R^2$ are taken from PredictDB ("test_R2_avg").
(TIFF)

**S11 Fig. A comparison of $R^2$ from SAGE and MESA_AFHI training diagnostics, similar to Fig 3.** The SAGE $R^2$ are computed from regressing PrediXcan predictions onto gene

expression measurements. The MESA_AFHI $R^2$ are taken from PredictDB ("test_R2_avg").
(TIFF)

**S12 Fig. A comparison of $R^2$ from SAGE and MESA_CAU training diagnostics, similar to Fig 3.** The SAGE $R^2$ are computed from regressing PrediXcan predictions onto gene expression measurements. The MESA_CAU $R^2$ are taken from PredictDB ("test_R2_avg").
(TIFF)

**S13 Fig. Violin plots of the distribution of $R^2$ from Table 1.** The train-test scenarios from left to right are Africans to Africans, Africans to non-Finnish Europeans, Africans to all Europeans, Africans to Finns, non-Finnish Europeans to Africans, non-Finnish Europeans to non-Finnish Europeans, non-Finnish Europeans to Finns, all Europeans to Africans, and all Europeans to all Europeans.
(TIFF)

**S14 Fig. Violin plots of the distribution of $R^2$ from Table 2.** The groups are ordered as in S13 Fig. The 564 genes presented here are somewhat better predicted than average. Effects of training sample size are evident, in which Europeans (EUR373, n = 373) generally yield higher $R^2$ than Africans (AFR, n = 89). A notable distributional different exists between Europeans predicting into Africans (EUR373 to AFR) vs. Europeans (EUR373 to EUR373), the latter of which shows an upward bias of $R^2$.
(TIFF)

**S15 Fig. Distributions of $R^2$ across all 25 train-test scenarios using the 5 constituent GEU-VADIS populations (CEU, FIN, GBR, TSI, and YRI).** The distributions correspond to Table 3. Distributions from scenarios where a population predicts into itself (CEU_CEU, FIN_FIN, GBR_GBR, TSI_TSI, or YRI_YRI) have noticeably fewer $R^2$ near 0, indicating improved prediction.
(TIFF)

**S16 Fig. Distributions of $R^2$ across 142 genes in common to all 25 train-test scenarios.** The train-test scenarios are ordered as in S15 Fig. The 142 genes represented here are predicted better than average. Population-level differences can be seen, particularly between the four European populations (CEU, FIN, GBR, and TSI) and YRI.
(TIFF)

**S17 Fig. Prediction $R^2$ between AFR (YRI) and EUR (CEU, TSI, GBR, and FIN).** Predicting into and from AFR produces consistently lower $R^2$ than predicting within EUR, suggesting a potential decrease in prediction accuracy when predicting across continental population groups.
(TIFF)

**S18 Fig. Genetic distance versus prediction accuracy over 142 genes with positive correlation across all train-test scenarios.** Here the GEUVADIS populations are arranged into three groups. AFR to AFR includes prediction from YRI into itself; EUR to AFR includes prediction into YRI from CEU, GBR, TSI, and FIN; and EUR to EUR includes prediction within and between all European populations in GEUVADIS. Clustering by genetic distance separates prediction between European populations from prediction between European populations and AFR. $F_{ST}$ are taken from the 1000 Genomes Project (S11 Table) (The 1000 Genomes Consortium, 2010).
(TIFF)

**S19 Fig. A schematic of three shared eQTL architectures for the case of k = 10 eQTLs per gene.** Blue encodes eQTLs specific to CEU; red encodes eQTLs specific to YRI; and gold

encodes eQTLs shared between CEU and YRI. Models for CEU and YRI always had k eQTLs. AA always inherited all eQTLs from the ancestral populations. Consequently, the number of eQTLs in AA varied depending on how many eQTLs CEU and YRI shared.
(TIFF)

**S20 Fig. Correlations between predictions and simulated gene expression measurements from simulated populations across various proportions of shared eQTL architecture with 20 causal cis-eQTLs.**
(TIFF)

**S21 Fig. Correlations between predictions and simulated gene expression measurements from simulated populations across various proportions of shared eQTL architecture with 40 causal cis-eQTLs.**
(TIFF)

**S22 Fig. Mean correlations between predictions and simulated gene expression measurements from simulated populations for a single causal cis-eQTL.** For this simplified eQTL architecture, the ancestral populations (CEU and YRI) either share the causal eQTL (TRUE) or not (FALSE). In the TRUE case, AA has 1 eQTL shared with CEU and YRI; in the FALSE case, it has 2 unique eQTLs, one from each of CEU and YRI. Error bars denote 95% confidence intervals.
(TIFF)

**S23 Fig. Distributions of t-statistics across various shared eQTL proportions for all nine train-test scenarios with 1000 Genomes populations for a fixed TWAS effect size and fixed number of causal eQTLs.** The labels are ordered from left to right from least admixture proportion (CEU to YRI, 0% admixture proportion) to highest admixture proportion (YRI to AA, 80% admixture proportion), with train-test scenarios from a population into itself on the right of each panel. Increasing proportions of shared eQTLs yield stronger association statistics from cross-population predictions. Fully shared eQTL architectures yield consistently high power across populations. Median t-statistics increase as populations share more haplotypes, while association tests with gene expression predicted in the same population show consistently high power.
(TIFF)

## Acknowledgments

The authors wish to acknowledge the following SAGE co-investigators for subject recruitment, sample processing and quality control: Luisa N. Borrell, DDS, PhD, Emerita Brigino-Buenaventura, MD, Adam Davis, MA, MPH, Michael A. LeNoir, MD, Kelley Meade, MD, Fred Lurmann, MS and Harold J. Farber, MD, MSPH. The authors also wish to thank the staff and participants who contributed to the SAGE study. A significant portion of this research was conducted at the Berkeley Institute for Data Science (BIDS). The logistical space, technical support, administrative assistance, and indefatigable good humor of the members and staff at BIDS is gratefully acknowledged.

## Author Contributions

**Conceptualization:** Kevin L. Keys, Christopher R. Gignoux.

**Data curation:** Angel C. Y. Mak, Jennifer R. Elhawary, Donglei Hu, Scott Huntsman, Sam S. Oh, Sandra Salazar, Michael A. Lenoir, Christopher R. Gignoux.

**Formal analysis:** Kevin L. Keys.

**Funding acquisition:** Esteban G. Burchard.

**Investigation:** Kevin L. Keys.

**Methodology:** Kevin L. Keys, Angel C. Y. Mak, Marquitta J. White, Walter L. Eckalbar, Andrew W. Dahl, Joel Mefford, Anna V. Mikhaylova, Jimmie C. Ye, Timothy A. Thornton, Noah Zaitlen, Christopher R. Gignoux.

**Project administration:** Kevin L. Keys, Angel C. Y. Mak, Sandra Salazar, Esteban G. Burchard, Christopher R. Gignoux.

**Resources:** Angel C. Y. Mak, Celeste Eng, Donglei Hu, Scott Huntsman, Sam S. Oh, Sandra Salazar, Michael A. Lenoir, Christopher R. Gignoux.

**Software:** Kevin L. Keys, Angel C. Y. Mak, Walter L. Eckalbar, Donglei Hu, Scott Huntsman, Christopher R. Gignoux.

**Supervision:** Kevin L. Keys, Timothy A. Thornton, Christopher R. Gignoux.

**Validation:** Kevin L. Keys.

**Visualization:** Kevin L. Keys, Angel C. Y. Mak, Marquitta J. White, Christopher R. Gignoux.

**Writing – original draft:** Kevin L. Keys, Christopher R. Gignoux.

**Writing – review & editing:** Kevin L. Keys, Angel C. Y. Mak, Marquitta J. White, Walter L. Eckalbar, Andrew W. Dahl, Joel Mefford, Anna V. Mikhaylova, María G. Contreras, Jennifer R. Elhawary, Celeste Eng, Donglei Hu, Scott Huntsman, Sam S. Oh, Sandra Salazar, Michael A. Lenoir, Jimmie C. Ye, Timothy A. Thornton, Noah Zaitlen, Esteban G. Burchard, Christopher R. Gignoux.

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
