## [Decision Letter · Decision Letter 0]

25 Dec 2019

Dear Dr Keys,

Thank you very much for submitting your Research Article entitled 'On the cross-population generalizability of gene expression prediction models' to PLOS Genetics. Your manuscript was fully evaluated at the editorial level and by independent peer reviewers. The reviewers appreciated the attention to an important problem, but raised some substantial concerns about the current manuscript. Based on the reviews, we will not be able to accept this version of the manuscript, but we would be willing to review again a much-revised version. We cannot, of course, promise publication at that time.

If you decide to revise the manuscript for further consideration at PLOS Genetics, please aim to resubmit within the next 60 days, unless it will take extra time to address the concerns of the reviewers, in which case we would appreciate an expected resubmission date by email to plosgenetics@plos.org.

[LINK]

We are sorry that we cannot be more positive about your manuscript at this stage. Please do not hesitate to contact us if you have any concerns or questions.

Yours sincerely,

Tuuli Lappalainen

Associate Editor

PLOS Genetics

Gregory Barsh

Editor-in-Chief

PLOS Genetics

Reviewer's Responses to Questions

**Comments to the Authors:**

Reviewer #1: Keys et al. use gene expression prediction models publicly available in PredictDB from multiple cohorts (DGN, GTEx, MESA) to predict gene expression in SAGE (n=39), an African American pediatric asthma cohort with genome-wide genotypes and whole blood RNA-Seq available. They assess predictive performance by comparing predicted expression to observed. They go on to build prediction models in GEUVADIS and test them within and between EUR and AFR populations within GEUVADIS. They also generate simulated African American data and show predictive performance and TWAS power increases with increased shared eQTL genetic architecture.

I have reviewed this paper previously at another journal and while the authors have added TWAS simulations and addressed most of my concerns, a major one remains. Since prediction performance in SAGE is poor across all PredictDB models tested, even among the best-predicted genes (Fig 3), is it the best cohort to use to address the question of cross-population generalizability, especially when larger datasets (GEUVADIS) are available? The authors should address whether a test set of 39 individuals is a suitable sample size for reliable estimates of performance. In addition to sample size, other confounders like age, population structure, and hidden confounders could also affect performance in SAGE. In the methods, you state that you followed the GTEx v6p eQTL QC pipeline, but details about how many PEER factors, genotypic PCs, etc. were used in your SAGE analyses are needed, especially given n=39 is much smaller than any GTEx eQTL tissue. By using just one small validation cohort in the first section of your paper (SAGE), key population performance differences among PredictDB models may be missed. I suggest using GEUVADIS populations as validation cohorts of PredictDB models to see if larger sample sizes reveal the expected differences among populations.

The GEUVADIS results as presented are stronger and a useful demonstration of the cross-population prediction problem observed by others (Mogil et al. 2018, Mikhaylova et al. 2019). The GEUVADIS prediction models built by the authors would be useful to the community and should be made publicly available. While summary stats (R2, rho) were available at https://ucsf.box.com/v/sage-geuvadis-predixcan, the prediction models (i.e. SNP weights per gene) were not included.

The simulations are thorough and well-presented, and the portion of this paper that goes beyond previous studies. Simulated population models with non-identical eQTLs showed patterns similar to real-world data. The authors call for more diverse sampling in transcriptome studies is timely and necessary in order better compare eQTL architecture among populations.

Minor:

1. Line 322. Change “different” to “differences”.

2. Line 404. Add “in” to “predicted these populations”.

3. I think the last sentence fragment in Figure 5 legend should be deleted.

4. Suppl. Fig. 5. Wide bar in the AA 50% circle should be yellow (shared), not red.

Reviewer #2: Cross population prediction of gene expression is an incredibly timely subject given that the bulk of match genotype-gene expression data has been done in European ancestry individuals and is use to predict gene expression in diverse population, without complete understanding of the disadvantages or potential errors that may occur. This paper highlights the need to create prediction models in populations that reflect the ancestry of genotypes used to impute. The authors have used both real and simulated data to make this case, and overall, I feel that paper is useful to the human genetic community. But the manuscript suffers from errors in data visualization that distract from the message and leave the reader confused as to which is the true representation of the data. Below are outline comments to improve and correct these errors.

Major Comments.

1. The number of gene models available in each weighted set would be highly dependent on the criteria used to identify these gene as “well predicted”. As an example, depending on the R2 threshold the MESA_All set could have anywhere from 6896 gene model to 1336 gene models (as per the paper). The authors need to add details to the methods on what criteria was used to get to the final list of gene model that are compared. Where these thresholds the same for all weighed sets?

2. By using all 11,545 genes in the analysis, it seems like it would bias the estimate downward. Especially since the authors compare these vales to the R2 from the training set which do not include all 11,545 genes in all models. While this point is somewhat answered by the GTExv7 analysis on well predicated genes, this does not address if this is true for the more diverse cohorts like MESA.

3. The box and whiskers plot are not particularly informative to show the difference in R2 between the weighted sets as highlighted in the results. While I understand the want to graph the test R2 with the actual R2, I think it may be easier to see when graphed separately.

4. As a follow up to comment 2, could the negative correlation described in results (lines 189 – 195) be due to the inclusion of all 11K genes as oppose to those well predicted by each model. Would this still be the case if the authors restricted to just those genes listed in Supplementary Table 1?

5. The results for the 564 genes that were trained in all GEUVADIS data should be presented similarly to the larger gene set (lines 241-247). Specifically, the author should highlight how the models trained on EUR fared in AFR and vise-versa as oppose to just giving ranges of the R2 across all comparisons.

6. I would be helpful to know the population sizes for each on the subpopulations presented on table 3 and 4, unless all of these are 89 as stated in line 252

7. I am a bit confused on the simulation in which the admixed population inherits causal eQTLs from the parental populations. The explanation in the results states that when k=10 and if only 50% of the alleles are shared, then the AA would have 15 causal eQTLs (5 from CEU, 5 from YRI and 5 shared). So, wouldn’t that be 15 causal eQTLs not 10? Also, would it not make more sense use all 5 shared and for the remaining 5 use 80% of the YRI specific and 20% of the CEU specific? While the authors explain the effect of this in the result it is hard to interpret how the simulation would reflect actual admixture.

8. I am confused by Figure 6 as the text and the figure do not seem to match. The authors states,” For an architecture with no shared eQTLs, power between CEU to YRI is 0, while power is higher for CEU to AA (0.25) and YRI to AA (0.30).” But the bar chart shows CEU to AA and YRI to AA both to be close to identical at around 0.8.

9. The matching of numbers reported in results to the figures continues on Figure 7 – though admittedly closer. The authors wrote, “For example, when h2 = 0.20 and gene expression is predicted from AD to CEU, power at 0% YRI admixture is 0.56 (95% CI: 0.462 – 0.658) and declines linearly with increasing YRI admixture; at 100% YRI, statistical power for AD to CEU is 0.46 (95% CI: 0.362 - 0.558).” but the line depicted is clearly at 0.5 when the YRI admixture is at 100%. The author need to thoroughly examine all figure to ensure they are depicting what is written in the text.

Minor comments:

1. In the simulated gene expression, the author chose cis-heritability of 0.15. Is this the average across all tissues in GTEx, or only LCL, as this may be tissue specific?

2. Why did the authors simulate the AA haplotypes as oppose to use the Hapmap ASW data and simulate haplotypes for that data?

3. 2 gene were removed from the TWAS analysis because they had no SNPs in the simulated data. I think this mean there were no SNPs within 2Mb of these genes. This strikes the reviewer as very odd. What is the explanation for this?

4. It would be useful to the reader to add the github repositories that contain the PredictDB models that were used in the paper.

5. Gene numbers do not agree between the results and the methods. As an example, the methods states that 10,161 gene were found in at least one weighed set. But in the result the number is 11,545.

6. The authors point out the least negative correlation was seen with MESA_AFHI and the most with MESA_AFA, but the number is the results are not shown on Supplementary Table 1.

7. “Utahans” is misspelled on line 217

8. Figure 5 is incomplete. The ledged reads, “A dotted red line at h2 = 0.95 marks the power values shown in”. Shouldn’t this be at 0.205 (assuming this is related to Figure 6) also there is no red line.

Reviewer #3: In this manuscript, the authors describe a impressive set of analyses on the portability of gene expression QTLs (eQTLs) across diverse ancestries. They then extend this work to look at Transcriptome-Wide Association Studies (TWAS), and evaluate the extent of portability of these models across ancestry groups.

In the case of complex traits, we have larger sample sizes and are often discouraged by the losses in portability there, but this paper clearly makes the important point that the situation is even more discouraging for gene expression. The paper is well written and all analyses seem thoughtfull laid out. In addition, there has been momentous effort to make the data and software freely available, which should be commended. That being said, there are a few limitations to the current analyses, detailed below.

Major concerns:

1) The data come from myriad studies of: whole blood, PBMCs, monocytes, and LCLs. There is not a direct comparison between sample types, despite the availability of GTEx models trained on both LCLs and whole blood. Adding such a comparison of trained LCL/WB models would help researchers apply the results. (The lack of direct comparison to monocytes is understandable given the very preliminary nature of the MESA whole blood RNA-seq data generated as part of TOPMed, and the lack of existing TWAS models. As such, lack of monocyte comparison I don't think is a concern.)

2) GTEx, particularly in v8, has a substantial number of African-American and Hispanic participants with LCLs and Whole Blood. If possible, these should be included as a separate group for evaluation, to test whether the claim regarding SAGE and MESA_AFA is generalizable to other datasets. There are numerous different factors which might be contributing to the lack of reproducibility (e.g. WGS vs genotyping arrays; age effects; RNA isolation and sequencing protocols; source material; etc) and including GTEx Whole Blood and LCL expression in HIS and AFA individuals as evaluation sets would enable direct evaluation of some of these factors.

Points of minor analysis:

3) It would be nice to know whether the 564 genes (Table 2), 142 genes (Table 4), or 273 genes (Supplementary Table 1) are representative of the whole transcriptome. A simple violin plot of the expression distribution of these genes and other genes, within each population, would suffice. It might be helpful to see their Test R^2 estimates as well, but I don't think that's critical.

4) Regarding the TWAS simulation: While the simulation itself only used 100 genes, I think that readers would also be interested in understanding the power were a whole genome expression panel were used. Is it possible to recompute significant thresholds and provide these "transcriptome-wide" significance estimates as well? It would also be useful to know what the expected variance explained by the genes is -- my understanding, for beta = 1, is that you are adding N(0, 0.1^2) noise still, so heritability should be very high? (It's a bit unclear whether the h^2 = 0.15 applies to the TWAS simulation as well, and lines 143-149 of simulate_twas_sge.R seem to indicate that h^2 isn't used.)

Points of clarification or minor analysis:

5) Given that the SAGE participants were ascertained on the basis of rs28450894, some validation that effect sizes are not out of line at this locus is important to understanding the results. How close are the 273 genes to this SNP, and does this SNP (or a close LD partner) have non-zero weight in any models? It does seem rather unlikely that this SNP (or bronchodilator status in general) are driving the lack of signal, though, so simply acknowledging this and noting these distances and weights is sufficient in my opinion.

6) It appears that there is now a HIS weight set for MESA available on predictdb.org -- I would suggest either including, or rewording "all four" (line 153) to "four of the".

7) Are you powered (or is it possible to obtain) to estimate the heritability of the 273 genes in each population? Evaluating whether there is a relationship between R^2 and h^2 would be valuable. Currently, the Test R^2 is the comparison group, which should still be included but is perhaps underpowered and should not be considered a true upper bound. In particular, it would be nice to know if h^2 is different in AFA than in EUR. At the very least, showing a scatterplot of Test R^2 in EUR vs AFA in MESA/GEUVADIS of the different gene sets would be helpful.

8) Adding to Table 1 the number of individuals in the train and test populations, as well as the number of genes with positive correlation, would help with direct interpretation of Table 1 and Table 2. In addition to the R^2 measures, perhaps a direct test of e.g. test statistic inflation be more interpretable?

Points of confusion or surprise in the text:

7) What does it mean for CEU and YRI to have 0.0 shared ancestry? I see the simulation used haplotypes from CEU, YRI, or a mix thereof, but these sets of haplotypes can overlap. Some measure of haplotype sharing would be appreciated (or just stating the simulation proportion mixing directly, without making a population-level claim).

8) Do you have a sense of why AA->AA is so much better than CEU->CEU or YRI->YRI in Supplemental Figure 9? Should I be interpreting this in the context of total haplotypic diversity in AA?

9) It seems from Supp Table 3 like the FIN testing population, trained in AFR, does substantially better than the other EUR test populations. Any idea why that might be?

10) For simulated gene expression that went into the TWAS, was there a minor allele frequency cutoff in choosing causal eQTLs? And was this matched across populations?

11) "The comparison of predictive models cannot easily differentiate predictions of 0 (no gene expression) and NA (missing expression) [L570-572]." -- Could you please clarify this statement?

Overall, this paper suggests (consistent with numerous other papers in the field) that prediction of binary and quantitative traits is not very portable across populations. This work extends that knowledge to gene expression, and importantly shows (through both real data and simulation) that the prediction accuracies are hindered as much or more than with complex traits.

**Have all data underlying the figures and results presented in the manuscript been provided?**

Reviewer #1: No: The GEUVADIS prediction models built by the authors would be useful to the community and should be made publicly available. While summary stats (R2, rho) were available at https://ucsf.box.com/v/sage-geuvadis-predixcan, the prediction models (i.e. SNP weights per gene) were not included.

Reviewer #2: Yes

Reviewer #3: Yes

PLOS authors have the option to publish the peer review history of their article (what does this mean?). If published, this will include your full peer review and any attached files.

Reviewer #1: No

Reviewer #2: No

Reviewer #3: No

---

## [Decision Letter · Decision Letter 1]

18 May 2020

Dear Dr Keys,

Thank you very much for submitting your Research Article entitled 'On the cross-population generalizability of gene expression prediction models' to PLOS Genetics. Your manuscript was fully evaluated at the editorial level and by independent peer reviewers. The reviewers appreciated the attention to an important topic but identified some minor aspects of the manuscript that should be improved. Please also respond to the reviewer's question about RNA-seq data access. 

We therefore ask you to modify the manuscript according to the review recommendations before we can consider your manuscript for acceptance. Your revisions should address the specific points made by each reviewer.

[LINK]

Yours sincerely,

Tuuli Lappalainen

Associate Editor

PLOS Genetics

Gregory Barsh

Editor-in-Chief

PLOS Genetics

Reviewer's Responses to Questions

**Comments to the Authors:**

Reviewer #1: The authors have addressed my concerns satisfactorily. The only minor adjustment in the text I suggest is removing/rewording this sentence from the Introduction, line 138: “To our knowledge, nobody has investigated cross-population generalizability of new prediction models generated within GEUVADIS.” Fryett et al. very recently (March 2020) published a study in Genetic Epidemiology that did this (https://doi.org/10.1002/gepi.22290). I understand their work was completed in parallel to yours, but I suggest removing this now inaccurate sentence. Doing so will in no way diminish your thorough investigation into the important problem of cross-population portability presented here. Thank you for your thoughtful response to the reviews and be well.

Reviewer #3: Thank you for your thoughtful and thorough response piece. I appreciate your inclusion of this additional material and I believe the resulting work is acceptable. A few brief comments:

1) It is worth noting that GTEx does have (self-report/close-relative-report) ethnicity in the ETHNCTY variable: https://ftp.ncbi.nlm.nih.gov/dbgap/studies/phs000424/phs000424.v8.p2/pheno_variable_summaries/phs000424.v8.pht002742.v8.p2.GTEx_Subject_Phenotypes.var_report.xml which I note because it might be helpful to include in future analyses. However, while I would still like to see the relative differences due to switching datasets versus switching cell types, I think the data are fine as they are. I would just request the authors make this limitation more clear, and that the predictions observed might change on other datasets from the same cell type versus on different cell types within whole blood.

2) In particular, I don't understand the argument that GTEx v8 and GTEx v7 are sufficiently different that they cannot be compared. Were that the case, would SAGE not also be too different to be compared? It might be worth noting cases under which whole blood could be predicted more accurately than in LCLs, e.g. https://pubmed.ncbi.nlm.nih.gov/19043577/.

3) In the abstract: "the amount of shared genotype predictors" is unclear and I would re-word to indicate that this refers to genetic variants included in the model.

4) The following response: "However, in light of the issues seen during our test, we believe that displaying the correlations is a more appropriate description and that there would be limited test statistic inflation." Suggests to me that including the test statistic inflation analysis would aid interpretation of the results. Alternatively, the fraction of FDR-adjusted positive correlations (under a half-normal distribution) could play a similar role. I think that everyone expects the power to be limited, but it is useful to have some measure of error on the R^2 measures. (for instance, with the current rendition it is unclear whether FIN is indeed better predicted than EUR278 with AFR weights)

5) Regarding your response to the FIN prediction, it suggests that if heterogeneity is driving differences in prediction, a meta-analysis across populations might be more appropriate. However I think the point is clear enough as is that such an analysis is likely above and beyond.

Otherwise, I think the manuscript is clear and comprehensive.

**Have all data underlying the figures and results presented in the manuscript been provided?**

Reviewer #1: No: SAGE RNA-Seq data did not appear to be available through dbGaP phs000921.v4.p1, just WGS. Please correct me if I'm wrong or provide an RNA-Seq accession or details on how the RNA data may be accessed.

Reviewer #3: Yes

PLOS authors have the option to publish the peer review history of their article (what does this mean?). If published, this will include your full peer review and any attached files.

Reviewer #1: No

Reviewer #3: No

---

## [Editor Report · Decision Letter 2]

10 Jun 2020

Dear Dr Keys,

We are pleased to inform you that your manuscript entitled "On the cross-population generalizability of gene expression prediction models" has been editorially accepted for publication in PLOS Genetics. Congratulations!

Yours sincerely,

Tuuli Lappalainen

Associate Editor

PLOS Genetics

Gregory Barsh

Editor-in-Chief

PLOS Genetics

Comments from the reviewers (if applicable):

**Data Deposition**

http://datadryad.org/submit?journalID=pgenetics&manu=PGENETICS-D-19-01922R2

**Press Queries**

---

## [Editor Report · Acceptance letter]

1 Jul 2020

PGENETICS-D-19-01922R2 

On the cross-population generalizability of gene expression prediction models 

Dear Dr Keys, 

We are pleased to inform you that your manuscript entitled "On the cross-population generalizability of gene expression prediction models" has been formally accepted for publication in PLOS Genetics! Your manuscript is now with our production department and you will be notified of the publication date in due course.

With kind regards,

Matt Lyles

PLOS Genetics

On behalf of:
